
# Evaluation on the effect of regional joint control measures in changing photochemical transformation: A comprehensive study of the optimization scenario analysis

Li LI[1,2#], Shuhui ZHU[2#], Jingyu AN[2#], Min ZHOU[2], Hongli WANG[2*], Rusha YAN[2], Liping QIAO[2],

Cheng Huang[2*], Xudong TIAN[3], Lijuan SHEN[4], Jeremy C AVISE[5], Joshua S FU[6]

1.  School of Environmental and Chemical Engineering, Shanghai University, Shanghai, 200444, China
2.  State Environmental Protection Key Laboratory of the Cause and Prevention of Urban Air Pollution Complex, Shanghai Academy of Environmental Sciences, Shanghai 200233, China
3.  Zhejiang Environmental Monitoring Center, Hangzhou, 310014, China
4.  Jiaxing Environmental Monitoring Station, Jiaxing, 314000, China
5.  Laboratory for Atmospheric Research, Washington State University, Pullman, Washington, USA.
6.  Department of Civil & Environmental Engineering, University of Tennessee, Knoxville, TN 37996, USA

*Correspondence to: C. Huang (huangc@saes.sh.cn) and H. L. WANG (wanghl@saes.sh.cn)

#These three people contribute equally.

**Abstract:** Heavy haze usually occurs in winter in eastern China. To control the severe air pollution during the season, comprehensive regional joint-control strategies were implemented throughout a campaign. To evaluate the effectiveness of these strategies and to provide some insight into strengthening the joint-control mechanism, the influence of control measures on levels of air pollution were estimated. To determine the influence of meteorological conditions, and the control measures on the air quality, in a comprehensive study, the 2nd World Internet Conference was held during December 16~18, 2015 in Jiaxing City, Zhejiang Province in the Yangtze River Delta (YRD) region. We first analyzed the air quality changes during four meteorological regimes; and then compared the air pollutant concentrations during days with stable meteorological conditions. Next, we did modeling scenarios to quantify the effects caused due to the air pollution control measures. We found that total emissions of $SO_2$, $NO_x$, $PM_{2.5}$ and VOCs in Jiaxing were reduced by 56%, 58%, 64% and 80%, respectively; while total emission reductions of $SO_2$, $NO_x$, $PM_{2.5}$ and VOCs over the YRD region are estimated to be 10%, 9%, 10% and 11%, respectively. Modelling results suggest that the regional controls (including Jiaxing and surrounding area) reduced $PM_{2.5}$ levels in Jiaxing between 5.5%-16.5% (9.9% on average), while local control measures contributed 4.5%-14.4%, with an average of 8.8%. Our implemented optimization analysis compared with previous studies also reveal that local emission reductions play a key role in air quality improvement, although it shall be supplemented by regional linkage. In terms of regional joint control, to implement pollution channel control 48 hours before the event is of most benefit in getting similar results. Therefore, it is recommended that a synergistic emission reduction plan between adjacent areas with local pollution emission reductions as the core part should be established and strengthened, and emission reduction plans for different types of pollution through a stronger regional linkage should be



reserved.
**Keywords:** PM$_{2.5}$; regional joint control; meteorology; YRD

**1 Introduction**

High concentrations of PM$_{2.5}$ has attracted much attention due to its impact on visibility (Pui
et al., 2014), human health (West et al., 2016) and global environment. To control air pollution
situation in China, the Ministry of Ecology and Environment of the People's Republic of China has
released a lot of policies, which can generally be divided into long-term action plans (such as the
Clean Air Action Plan (2013-2017), the Five-year Action Plans) and short-term control measures
(such as Clean Air Protection at Mega Events, Air Pollution Warning and Protection Measures).
China has successfully implemented some mega event air pollution control plans and ensured good
air quality, including the 2008 Beijing Olympics (Kelly and Zhu, 2016); the 2010 World Expo in
Shanghai (CAI-Asia, 2010); the 2010 Guangzhou Asian Games (Liu et al., 2013); the 2014 Asia-
Pacific Economic Cooperation Forum (APEC) (Liang et al., 2017); 2014 Summer Youth Olympics
in Nanjing (CAI-Asia, 2014) and the 2015 China Victory Day Parade (Victory Parade 2015) (Liang
et al., 2017), etc. After implementation of these control measures, it is important to understand how
effective these strategies are.
The 2$^{nd}$ World Internet Conference was held in Wuzhen, Jiaxing, Zhejiang during 16-18
December, 2015. To reduce air pollution during the conference, Zhejiang Province and the Regional
Air-pollution Joint Control Office of the Yangtze River Delta (YRD) region developed an Action
Plan for Air Pollution Control during the Conference (henceforth referred to as the Action Plan),
which clarified target goals, time periods for implementing controls, regions in which the controls
would be applied, and the control measures to be implemented, as described below. **Targets:**
achieve an Air Quality Index (AQI) below 100 in "key areas", an AQI below 150 in "control areas",
and to achieve significant improvement of the air quality in the surrounding (or buffer) regions
outside of the control areas. **Time Periods:** the time periods of interest for implementing various
controls include the early stage (3 months before the conference), the advanced stage (2 weeks to 4
days before the conference) and the central stage (3 days before and 2 days after the conference).
**Regions:** areas within a 50km radius, within a 100km radius and outside of a 100km radius from
the centre of Wuzhen were classified as key areas, control areas and buffer areas, respectively. These



areas cover 9 cities including Jiaxing, Huzhou, Hangzhou, Ningbo and Shaoxing in Zhejiang
province, Suzhou and Wuxi in Jiangsu province and Xuancheng in Anhui province.

Many studies have provided descriptive analysis of the changing concentrations of air

pollutants during mega events. Some have reported the emission reductions and related air quality
changes. However, different air pollution control targets, different control measures, and different
locations, may cause big different effects among those strategies. In this paper, the reduction in
PM$_{2.5}$ achieved through the Action Plan is investigated further to help quantify the level of PM$_{2.5}$
reduction that can be attributed to different aspects of the Action Plan. An integrated emission-
measurement-modelling method described in the next section including analysis of multi-pollutant
observations, backward trajectory and potential source contribution analyses, estimates of pollutant
emission reductions, and photochemical model simulations were adopted to conduct a
comprehensive assessment of the impact of control measures on air quality improvement based on
three aspects: meteorological conditions, pollutant emission reductions of local sources, and
regional contributions.
**2 Methodology**

In order to strengthen the regional air pollution joint-control mechanism in the YRD region,

various measures and their implementation were systematically reviewed, and the qualitative and
quantitative relationships between the implementation of measures, changes in emissions of air
pollution sources and air quality improvement were studied. Specifically, the impact of measures
such as management and control of coal-burning power plants, production restriction and
suspension of industrial enterprises, motor vehicle limitation and work site suspension, were
investigated. In addition, the role of meteorology (in particular, transport) was assessed in terms of
its influence on the relevance and effectiveness of various measures, and ways of optimising air
quality control measures and emergency emission reductions under heavy pollution during major
events were evaluated.

To assess the effectiveness of the various controls outlined in the Action Plan, emission

reductions associated with those controls were calculated, and photochemical modelling was
conducted to determine the change in PM$_{2.5}$ attributed to specific controls. On this basis, an
assessment of how to optimise control measures was carried out with respect to both the area in



which the emission reduction took place, as well as the start time for implementing the controls (i.e.,
how far in advance do the controls need to be implemented). Analysis of the numerical modelling
results is focused on the effectiveness of the control measures with respect to regional transport of
pollutants in the YRD region.
**2.1 Measurements**
The online observational station was set up at the Shanxi supersite of Zhejiang Province (30.82
N, 120.87 E), which was located at the core area for pollution-control measures. On-line hourly
$PM_{2.5}$ mass concentration, carbonaceous aerosols, elements, and ionic species were measured by the
Synchronized Hybrid Ambient Real-time Particulate Monitor (SHARP, model 5030, Thermo Fisher
Scientific Corporation, USA), the OC/EC carbon aerosol analyzer (Model-4, Sunset Laboratory
Corporation, USA), the Xact multi-metals monitor (XactTM 625, PALL Corporation, USA), and
the Ambient Ion Monitor-Ion Chromatograph (AIM IC, model URG 9000, URG Corporation, USA),
respectively. Meteorological parameters, consisting of wind speed, wind direction, temperature,
pressure, and relative humidity, were measured as well.
$PM_{2.5}$ concentration data conform to the standards of data quality control published by Ministry
of Ecology and Environment of the People's Republic of China.
A semi-continuous Sunset OC/EC analyser was used to measure OC and EC mass loadings at
the observation site by adopting NIOSH-5040 protocol based on thermal-optical transmittance
(TOT). The ambient air was first sampled into a $PM_{2.5}$ cyclone inlet with a flow rate of 8 L·$min^{-1}$.
The OC and EC were collected on a quartz fiber filter with an effective collection area of 1.13 $cm^2$.
The analyzer was programmed to collect aerosol for 45 min at the start of each hour, followed by
the analysis of carbonaceous species during the remainder of the hour. The analysis procedure is
described in detail by Huang et al. (2018)
The ionic concentrations of nitrate, sulphate, chloride, sodium, ammonium, potassium, calcium
and magnesium ($Na^+$, $K^+$, $Ca^{2+}$, $NH_4^+$, $Mg^{2+}$, $NO_3^-$, $SO_4^{2-}$, $Cl^-$) in the fine fraction ($PM_{2.5}$) were
measured with a 1-hour time resolution using the AIM IC. The sample analysis unit is composed by
an anion and a cation ion chromatographs (Dionex ICS-1100), which was using guard columns with
potassium hydroxide eluent (KOH) for the anion system and methane sulfonic acid (MSA) eluent
for the cation system. The limit of the detection reported by the manufacturer is 0.1 ug/$m^3$ for all





species. The operation principle of AIM-IC is described in detail by Markovic et al. (2012)
Hourly ambient mass concentrations of sixteen elements (K, Ca, V, Mn, Fe, As, Se, Cd, Au,
Pb, Cr, Ni, Cu, Zn, Ag, Ba) in $PM_{2.5}$ were determined by the Xact multi-metals monitor. In brief,
the Xact instrument samples the air through a section of filter tape at a flow rate of 16.7 lpm using
a $PM_{2.5}$ sharp cut cyclone. The exposed filter tape spot then advances into an analysis area where
the collected $PM_{2.5}$ is analyzed by energy-dispersive X-ray fluorescence (XRF) to determine metal
mass concentrations. The sequence of sampling and analysis were performed continuously and
simultaneously on an hourly basis.
**2.2 Potential Source Contribution Analysis**
TrajStat is a HYSPLIT model developed by Chinese Academy of Meteorological Sciences and
NOAA Air Resources Laboratory based on geographic information system (GIS). It uses statistical
methods to analyze air mass back trajectories to cluster trajectories and compute potential source
contribution function (PSCF) with observation data and meteorological data included (Wang et al.,

2009).

PSCF analysis is a conditional probability function using air mass trajectories to locate
pollution sources. It can be calculated for each 1° longitude by 1° latitude cell by dividing the
number of trajectory endpoints that correspond to samples with factor scores or pollutant
concentrations greater than specified values by the number of total endpoints in the cell (Zeng et al.,
1989). Therefore, pollution source areas are indicated by high PSCF values. Since the deviation of
PSCF results could increase with the raise of distance between cell and receptor, therefore a weight
factor (Wij) was adopted in this study to lower the uncertainty of PSCF results. PSCF and Wij
calculations are described in Eq. (1) and Eq. (2), where mij is the number of trajectory endpoints
greater than specified values in cell (i, j), nij is the number of total endpoints in this cell (Zeng et al.,
1989; Polissar et al., 1999).
$P = \dfrac{m_{ij}}{n_{ij}} \cdot W(n_{ij})$          (1)
$W(n_{ij}) = \begin{cases} 1.00, & 80 < n_{ij} \\ 0.70, & 20 < n_{ij} \le 80 \\ 0.42, & 10 < n_{ij} \le 20 \\ 0.05, & n_{ij} \le 10 \end{cases}$          (2)
In this study, the TrajStat modelling system was used to analyze potential source contribution



areas of PM2.5 in Jiaxing during different pollution episodes with the combination of GDAS
meteorological data provided by the NCEP (National Center for Environmental Prediction).
Pollution trajectories corresponded to those trajectories with PM2.5 hourly concentration higher
than 75 μg/m$^3$.

**2.3 Model setup for separating meteorological influence and control measures**

**2.3.1 Model selection and parameter settings**

In this study, the WRF-CMAQ/CAMx air quality numerical modelling system was used to
evaluate the improvement in air quality resulting from the control measures outlined in the Action
Plan. It takes into account of modeling variations from different air quality models. For the
mesoscale meteorological field, we adopted the WRF model Version 3.4
(https://www.mmm.ucar.edu/wrf-model-general), the CAMx model Version 6.1
(http://www.camx.com/) and the CMAQ model Version 5.0 (Nolte et al., 2015;
http://www.cmascenter.org/cmaq/). The chemical mechanism utilized in CMAQ was the CB05 gas
phase chemical mechanism (Yarwood, et al., 2005) and AERO5 aerosol mechanism, which includes
the inorganic aerosol thermodynamic model ISORROPIA (Nenes, et al., 1998) and updated SOA
yield parameterizations. The gaseous and aerosol modules used in CAMx are the CB05 chemical
mechanism and CF module, respectively. The aqueous-phase chemistry for both models is based on
the updated mechanism of the Regional Acid Deposition Model (RADM) (Chang et al., 1987).
Particulate Source Apportionment Technology (PSAT) coupled in the CAMx is applied to quantify
the regional contributions to PM$_{2.5}$ as well. The WRF meteorological modeling domain consists of
three nested Lambert projection grids of 36km-12km-4km, with 3 grids larger than the
CMAQ/CAMx modeling domain at each boundary. WRF was run simultaneously for the three
nested domains with two-way feedback between the parent and the nest grids. Both the three
domains utilized 27 vertical sigma layers with the top layer at 100hpa, and the major physics options
for each domain listed in Table 1. For the CMAQ/CAMx modelling domain shown in Figure 1, we
adopted a 36-12-4km nested domain structure with 14 vertical layers, which were derived from the
WRF 27 layers. The two outer domains cover much of eastern Asia and eastern China, respectively,
while the innermost domain covers the YRD region. The simulation period was from 1-18
December, 2015, during which 1-7 December was utilized for model spin-up and 8-18 December


was the key period for analysis of the modelling results with control measures.

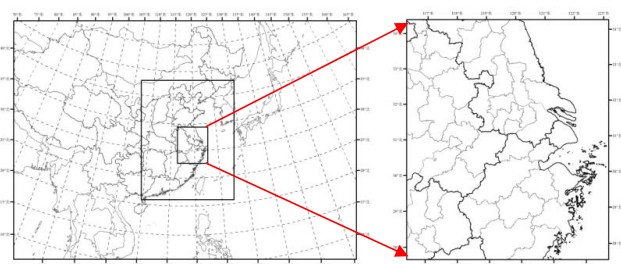

Fig 1. Modeling domain

Table 1 Parameterization scheme of the physical processes in the WRF model

| Physical Processes | Parameterization Scheme | Reference |
|---|---|---|
| Microphysical Process | Purdue Lin Scheme | (Lin, 1983) |
| Cumulus Convective Scheme | Grell-3 Scheme | (Grell and Dévényi, 2002) |
| Road Process Scheme | Noah Scheme | (Ek, 2003) |
| Boundary Layer Scheme | Yonsei University (YSU) Scheme | (Hong, 2006) |
| Long-wave Radiation | RRTM Long-wave Radiation Scheme | (Mlawer et al., 1997) |
| Short-wave Radiation Scheme | Goddard Short-wave Radiation Scheme | (Chou and Suarez, 1999) |

Initial and boundary conditions (IC/BCs) for the WRF modeling were based on 1-degree by 1-

degree grids FNL Operational Global Analysis data that are archived at the Global Data
Assimilation System (GDAS). Boundary conditions to WRF were updated at 6-hour intervals for
D01.

Anthropogenic source emission inventory in YRD is based on a most recent inventory

developed by our group (Huang et al., 2011;Li et al., 2011;Liu et al., 2018). The emission inventory
for areas outside YRD in China is derived from the MEIC model (Multi-resolution Emission
Inventory of China, latest data for 2012(http://www.meicmodel.org) and anthropogenic emissions
over other Asian region are from the MIX emission inventory for 2010 (Li et al., 2017). Biogenic
emissions are calculated by the MEGAN v2.1 (Guenther et al., 2012). The Sparse Matrix Operator
Kernel Emissions (SMOKE, https://www.cmascenter.org/smoke) model is applied to process these
emissions for modeling inputs that is more detailed emission processes and not usually used in China.
**2.3.2 Model performance**

Prior to evaluating the effectiveness of the control measures and reactions, the performance of




the modelling system was evaluated to ensure it was able to reasonably reproduce the observed
meteorological conditions and PM$_{2.5}$ levels. Statistical indexes used for model evaluation include
Normalised Mean Bias (NMB), Normalised Mean Error (NME) and Index of Agreement (I).

Observational data from the Shanxi supersite in Jiaxing City were compared with model results

for model evaluation verification. Table 2 shows the summary statistics for the main meteorological
parameters simulated with the WRF model and hourly PM$_{2.5}$ concentrations simulated by CMAQ.
Among the meteorological parameters, wind speed is slightly over predicted with the IOA value of
28%, while temperature, relative humidity and pressure all have NMB values greater than 0.9.
Figure 2 compares the simulated and observed PM$_{2.5}$ concentrations at the Shanxi supersite. In
general, model predicted data are lower than the observed data with the NMB value of -22% to -
30%, the NME value of 45% to 47% and the I value of 0.67 to 0.70 (Table 2). Overall, these statistics
for both the meteorological parameters and simulated PM$_{2.5}$ are generally consistent with the results
in other published modelling studies(Zheng et al., 2015;Wang et al., 2014;Zhang et al., 2011;Fu et
al., 2016;Li et al., 2015b;Li et al., 2015a), which suggests that the simulation performance is
acceptable.
Table 2 Statistics of simulation verification for meteorological parameters and hourly PM$_{2.5}$ concentration

| Statistical indexes | Wind speed | Temperature | Relative humidity | Air pressure | CAMx-PM$_{2.5}$ | CMAQ-PM$_{2.5}$ |
|---|---|---|---|---|---|---|
| NMB | 28% | 3% | -9% | 0% | -30% | -22% |
| NME | 33% | 14% | 12% | 0% | 45% | 47% |
| IOA | 0.81 | 0.97 | 0.93 | 1.00 | 0.67 | 0.70 |



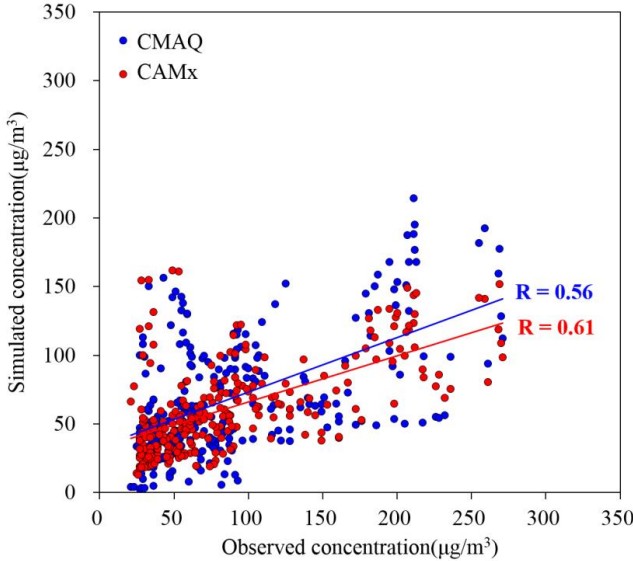

Fig. 2 Scatter plot of the simulated and observed PM$_{2.5}$ at the Shanxi supersite

**2.3.3 Method for quantifying the effectiveness of a control**

Quantifying the PM$_{2.5}$ reduction in response to emission reductions was done using the so

called Brute Force Method (BFM) (Burr and Zhang, 2011), where a baseline scenario was simulated
using unadjusted emissions (i.e., those emissions that would have occurred in absence of the Action
Plan) and a campaign scenario was modelled based on the emission controls outlined in the Action
Plan. In both cases, the same meteorology and chemical boundary conditions were utilized to drive
the photochemical model simulations. Through a comparative analysis of the scenarios, a relative
improvement factor (RF) for a given atmospheric pollutant, resulting from emission controls, can
be calculated and combined with ground based observations to assess the improvement in air quality
associated with those emission controls.

$RF = (C_b - C_s)/C_b$             (3)

$C_d = C_o \cdot RF$                (4)

where C$_b$ is the simulated pollutant concentration in the baseline scenario (µg/m$^3$), C$_s$ is the

pollutant concentration in the campaign scenario (µg/m$^3$), C$_o$ denotes the actual observed
concentration at the site (µg/m$^3$) and C$_d$ is the concentration improvement caused by the control
measures (µg/m$^3$). Utilizing models in a relative sense to assess the efficacy of emission controls on
air quality is common practice in regulatory modelling, with the assumption that there may be biases
in the absolute concentrations simulated by a modelling system, but that the relative response of that



system will reflect the response observed in the atmosphere (US EPA, 2014).
**3 Results and discussion**
**3.1 Photochemical transformation changes of air pollutants during the campaign**
Ground observation data show that from December 1 to December 23, Jiaxing City
experienced four distinct physical and chemical processes that contributed to the observed pollution
levels at different times. For each of these processes, this study has comprehensively in the
integrated emission-measurement-modeling method considered the backward air flow trajectory,
potential contribution source areas, meteorological conditions and the variation of $PM_{2.5}$
concentration to analyse the evolution of the observed air quality.
**3.1.1 Pollution process before the campaign with local emission accumulation as the main**
**contributor**
The first time period of interest was from December 6 to December 8. Analysis about the
potential source contribution areas resulting from PSCF modelling suggests that the polluted air
mass primarily originated from the northwest and northerly airstreams, passing Shandong, the
eastern coastal areas of Jiangsu and Shanghai and into northern Zhejiang, as is shown in Fig. 3.
Analysis of the large-scale weather patterns showed that pollution occurred in Beijing, Tianjin,
Shandong peninsula and northern Jiangsu as a result of cold air with polluted air mass transported
into the region on the morning of December 5. In the southern part of Shandong province, the $PM_{2.5}$
concentration peak appeared on the morning of December 6, while the $PM_{2.5}$ concentration peak
appeared around midnight on December 7 at the coastal area of Jiangsu. On December 6, the
development of warm and humid air flow, resulted in increasing ground humidity, which
contributed to the growth of secondary fine particles and the gradual accumulation of pollution in
northern Zhejiang and the surrounding areas of Shanghai. On December 7, affected by the surface
high-pressure system, the spread of pollution was slow, and the spatial extent of the pollution in
northern Zhejiang expanded. Therefore, during this time period, the pollution was primarily affected
by regional transport and worsened by stagnant local conditions in Jiaxing.



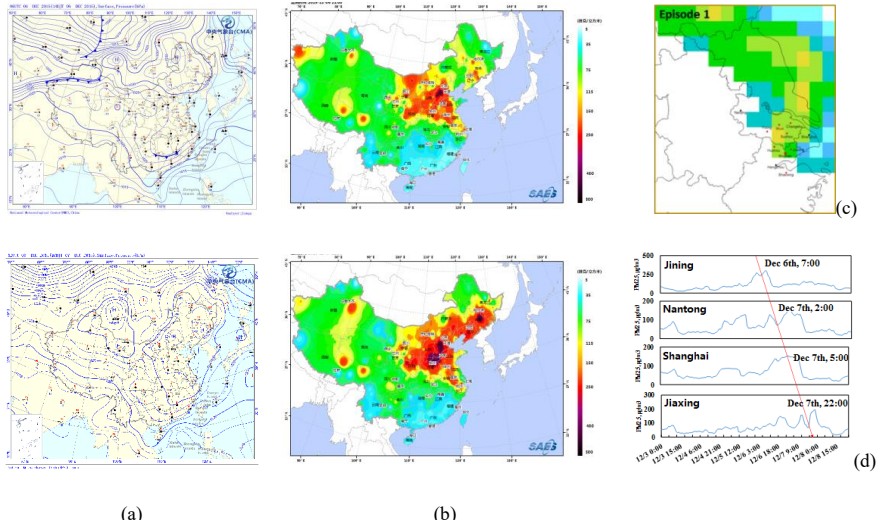

Fig. 3 Analysis of (a) the large-scale weather patterns, (b) distribution of PM$_{2.5}$ concentrations, (c) potential regional
sources, (d) PM$_{2.5}$ time series for selected sites during December 6 to December 8, 2015





### 3.1.2 Pollution process during the campaign with the southward motion of the weak cold air

The second time period of interest was from December 10 to December 11. Analysis about potential source contribution areas suggests that the polluted air mass mainly came from northern regions, passing from south-eastern Shandong peninsula and central-eastern Jiangsu to northern Zhejiang. From the large-scale weather pattern, the diffusion of weak cold air on December 10 gradually transported the pollution in the upper reaches of the region to the YRD region. The pollution peaked in areas such as Lianyungang in northern Jiangsu on the evening of December 10. On December 11, the PM$_{2.5}$ concentration peak appeared in central and southern Jiangsu as a result of northern weak air flow. The pollution was further transported into Zhejiang province with the expansion in influenced areas as is shown in Figure 5. Therefore, the pollution process was mainly affected by the transportation of polluted air mass caused by the southward motion of cold air.

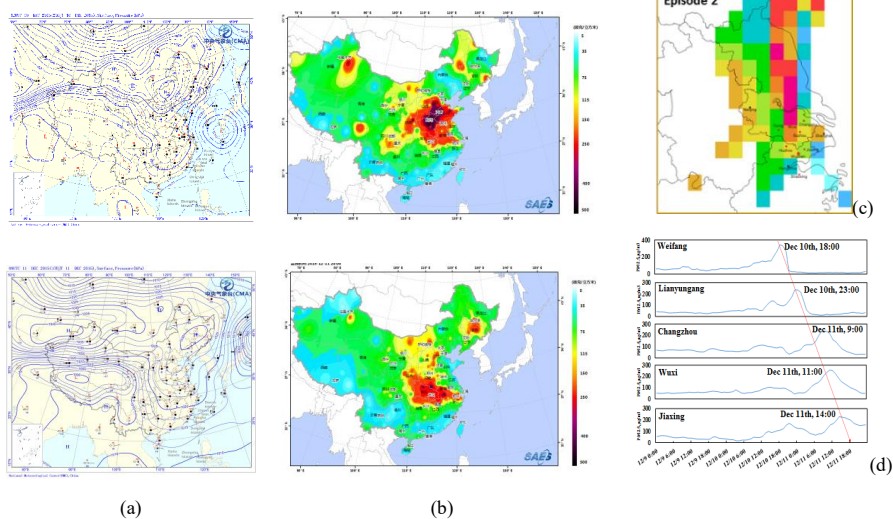

(a)  (b)

Fig. 4 Analysis of (a) the large-scale weather patterns, (b) distribution of PM$_{2.5}$ concentrations, (c) potential regional sources, (d) PM$_{2.5}$ time series for select sites during December 10 to December 11, 2015

### 3.1.2 Heavy pollution process during the campaign with the transit and transportation of strong cold air

The third period of interest was from December 13 to the early hours of December 16. Analysis of the potential source contribution areas suggest that the polluted air mass mainly came from the northwest direction, passing through south-eastern Shanxi, western Shandong, eastern Anhui and western Jiangsu to Zhejiang province. On December 14, affected by the cold air transportation in the north, northern pollution hit Hebei, Henan and Anhui provinces, with the highest degree of pollution on the 14th. On December 15, the further spread of cold air caused the transport of pollution into Jiangsu and Zhejiang. The northern part of Zhejiang province was in the centre of pollution on the 15th, which worsened the pollution and expanded the scope of pollution, as is shown in Figure 3-12. On December 16, under the control of the high-pressure system in northern Zhejiang, the air mass gradually





moved eastward and the air quality improved in the morning. Therefore, for this time period, large-scale
transportation was the main factor leading to the increase in pollutant levels.

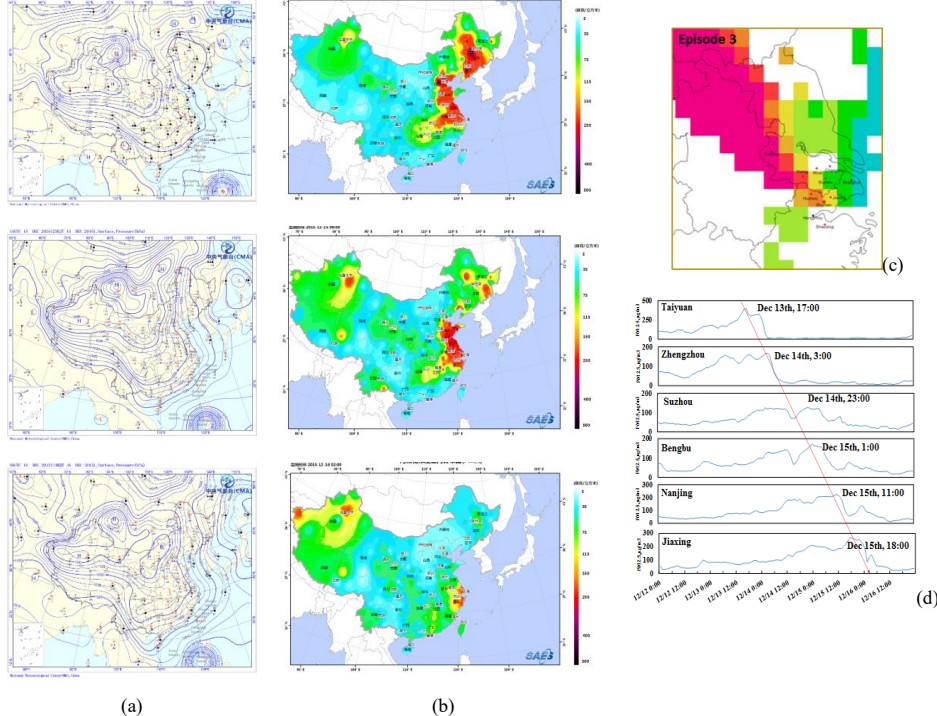

(a)                    (b)

Fig. 5 Analysis of (a) the large-scale weather patterns, (b) distribution of PM$_{2.5}$ concentrations, (c) potential regional sources, (d)
PM$_{2.5}$ time series for select sites during December 14 to December 16, 2015

**3.1.3 Pollution removal process caused by clean cold air during the conference**

During the conference from December 16 to December 18, weather was affected by the large-scale southward

transport of cold dry air in northern Zhejiang, resulting in lower temperature and relative humidity, as well as a
significant improvement in the air quality. On the 17th and the 18th, under the control of a high pressure system in
northern Zhejiang, the sea level pressure increased, the humidity was lower and the wind speed was reduced.
Because of the emission reduction effect of the control measures, the pollutant accumulation rate was likely slowed
and the air quality in northern Zhejiang was good overall. From the analysis of potential sources, PM$_{2.5}$
concentrations in Shandong, Jiangsu and Shanghai were significantly reduced. The PM$_{2.5}$ concentration during the
conference was mainly controlled by local emissions, as is shown in Figure 6.




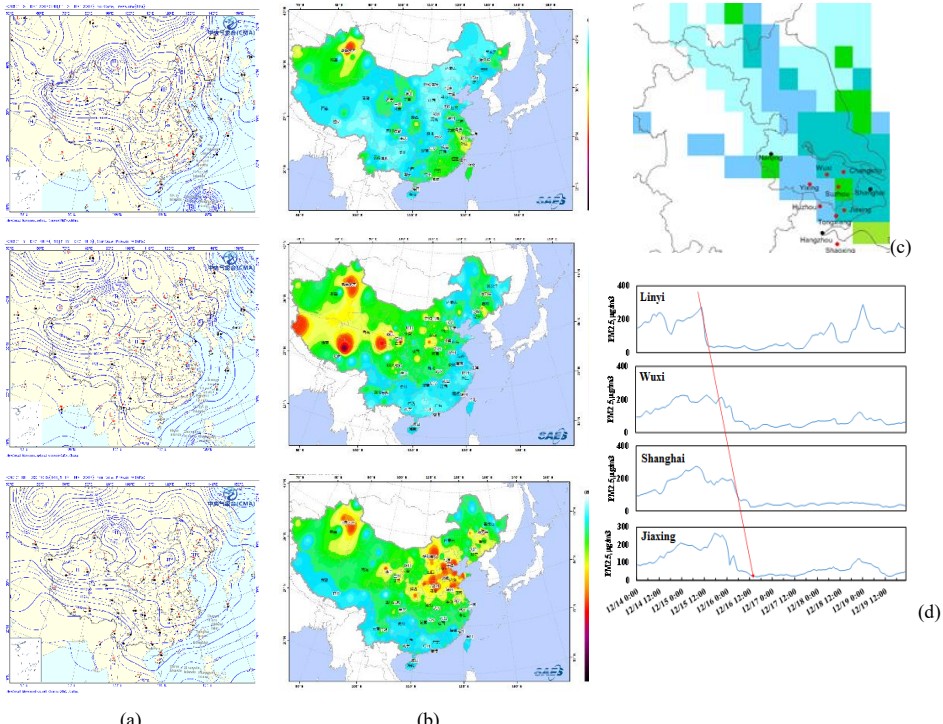

Fig. 6 Analysis of (a) the large-scale weather patterns, (b) distribution of PM$_{2.5}$ concentrations, (c) potential regional sources, (d)
PM$_{2.5}$ time series for select sites during December 16 to December 18, 2015
**3.1.4 Pollution process after the campaign with local emission accumulation as the main contributor**

The fourth period of interest was from December 20 to December 23. Analysis of the potential source

contribution areas suggest that the polluted air mass mainly came from the southwest direction, passing through
southern Hubei, southern Anhui and south-western Jiangsu to northern Zhejiang. On December 20, controlled by a
stagnant air mass, Zhejiang province has a relatively low near-surface wind speed and little dispersion, resulting in
the accumulation of local pollutants. On December 21, northern Zhejiang was located in the centre of a high pressure
system with conditions conducive to little mixing, and therefore pollution occurred in some areas in northern
Zhejiang. On December 22, affected by the warm and humid southwest air flow, Zhejiang had experienced some
precipitation but the pollution in northern Zhejiang was not improved due to deep polluted air masses. In Hubei and
Anhui located in the southwest of Jiaxing City, high pollution levels appeared from the evening of December 22 to
the early hours of December 23 as is shown in Figure 7. On December 23, the further expansion of polluted air
masses resulted in serious pollution in Jiangsu and northern Zhejiang. In general, under these heavily polluted
conditions, the local accumulation of pollutants was mainly caused by stagnant conditions with little dispersion and
transport within southwest air stream.





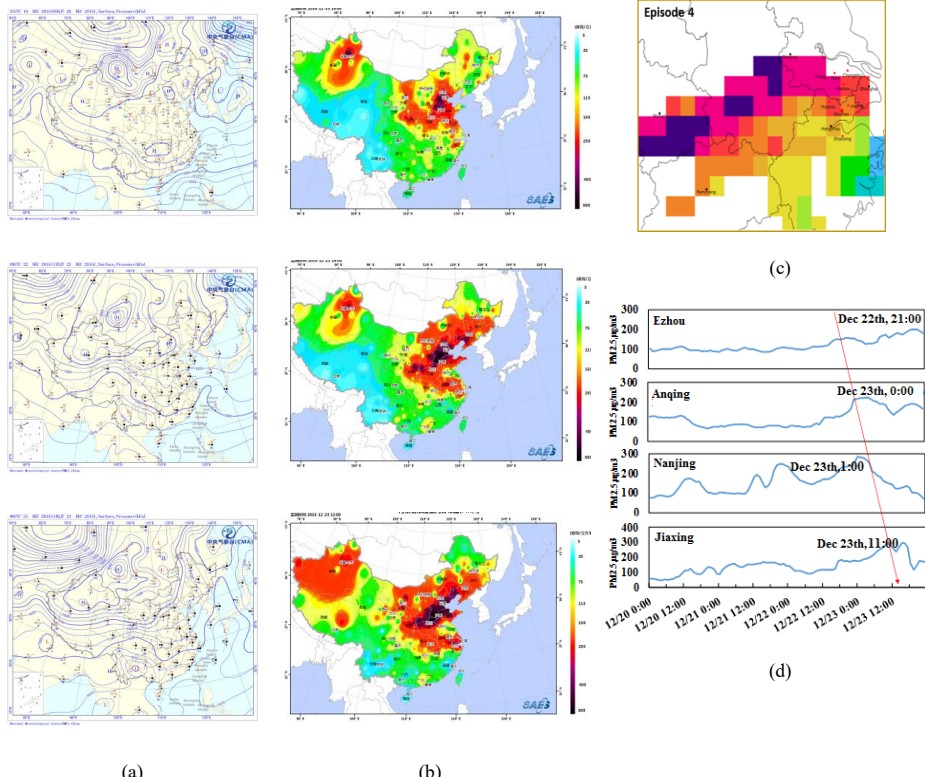

Fig. 7 Analysis of (a) the large-scale weather patterns, (b) distribution of PM$_{2.5}$ concentrations, (c) potential regional sources, (d)
PM$_{2.5}$ time series for select sites during December 20 to December 23

**3.2 Air quality changes under the same meteorological conditions before and after the campaign**

**3.2.1 Air quality changes under static meteorological conditions before and during the campaign**

During the air pollution control campaign for the conference, air quality in Jiaxing City fluctuated greatly due to the frequent southward motion of cold air from the north. Under static weather conditions, sources of atmospheric pollution mainly came from the accumulation of pollution from local sources and sources in neighbouring areas. Therefore, in order to eliminate the influence of the transportation process of the air mass, this study compared the air quality status before, during and after the campaign in Jiaxing City under stagnant weather conditions (wind speed less than 1m/s) and assessed the impact of control measures on ambient air quality in Jiaxing based on air quality observation data.

Figure 8 shows the concentration levels of normal pollutants including SO$_2$, NO, CO, NO$_2$ and PM$_{2.5}$ in Jiaxing City before (December 1-7), during (December 8-19) and after the campaign (December 19-31) under stagnant weather conditions. It can be seen that pollutant concentrations during the campaign were less than those before the campaign, in which SO$_2$ had the most significant decline of 40.1%, NOx, CO, PM$_{2.5}$ and PM$_{10}$ declined 8.0%, 2.6%, 12.5% and 16.3%, respectively, indicating that control measures have significantly improved the air quality in





Jiaxing City, especially with respect to $SO_2$ and $PM_{10}$.
After the campaign, all the pollutant concentrations rebounded sharply. $SO_2$, NO, $NO_2$, CO, $PM_{2.5}$, $PM_{10}$
increased 8.3%, 15.4%, 10.3%, 31.8%, 32.2% and 28.6%, respectively. Concentrations of some pollutants were
even higher than those before the campaign, which suggests that the emission intensity of the sources had
significantly increased after the campaign.

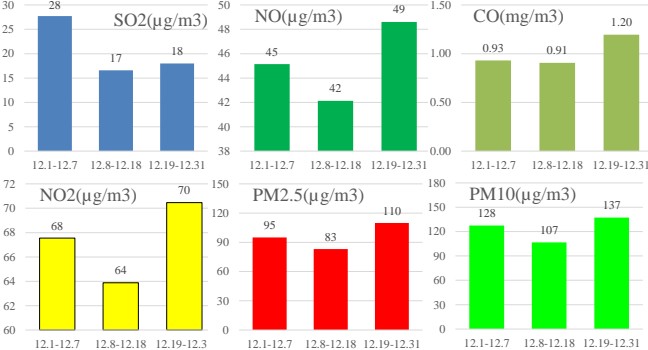

Fig. 8 Comparison between air pollutant concentrations at Shanxi station before, during, and after the campaign under stagnant
meteorological conditions
There are also some differences in concentrations of major chemical components of $PM_{2.5}$ in Jiaxing City
before (December 1-7), during (December 8-19) and after the campaign (December 19-31) under stagnant weather
conditions, as shown in Figure 9. The concentrations of major chemical components of $PM_{2.5}$ during the campaign
were less than those before the campaign, which is consistent with the conclusion about changes in normal pollutant
concentrations. On average, $SO_4^{2-}$, $NH_4^+$, $NO_3^-$, OC mineral soluble irons ($Ca^{2+}$ and $Mg^{2+}$) and $K^+$ declined 11.8%,
5.1%, 32.1%, 9.8%, 56.8% and 5.1%, respectively. Comparisons between the distribution of $PM_{2.5}$ chemical
components before and during the campaign suggest that $Ca^{2+}$ and $Mg^{2+}$ decreased most significantly during the
control period, which indicates that the suspension of construction operations which result in dust emissions and
the rising frequency of rinsing and cleaning paved roads, significantly reduced dust emissions. During the campaign,
$NO_3^-$ significantly decreased, indicating that vehicle control measures successfully reduced $NO_x$ emissions and
subsequently the formation of inorganic aerosols. The significant decrease in $SO_4^{2-}$ also shows that restricting and/or
suspending the operation of coal-burning power plants and industries in local and neighbouring cities played a very
positive role.



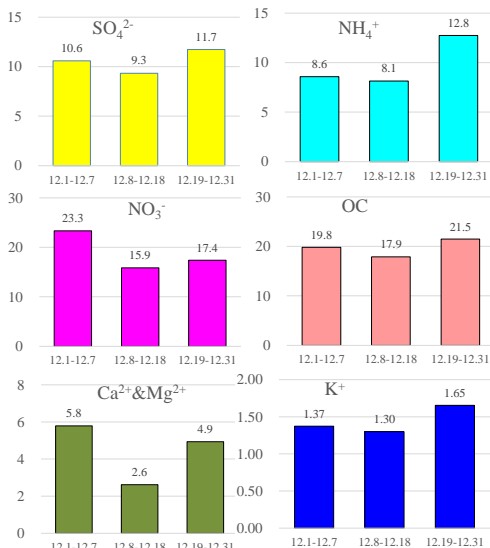

Fig. 9 Comparison between PM$_{2.5}$ chemical components at Shanxi station before and after the campaign under static meteorological
conditions

**3.2.2 Air quality changes under the same air mass trajectory before and during the campaign**

In order to distinguish the impact of meteorological conditions on air quality in Jiaxing City and better analyse
the effects of control measures on air quality during the conference, this study has combined meteorological
conditions with backward air flow trajectory analysis and carried out a comparative study by selecting a relatively
similar pollution period before and during the campaign. The first period occurred before the campaign from 12:00
December 2 to 20:00 December 4, while the second period occurred during the campaign from 9:00 December 16
to 5:00 December 18. Both of these periods were relatively unaffected by long-range transport of pollution into the
study area, and have similar backward airflow trajectories and meteorological conditions. Table 3 and Figure 10
compare average mass concentrations of pollutants (SO$_2$, NO$_x$, PM$_{2.5}$ and PM$_{10}$) during these two periods. As can
be seen from the figure, SO$_2$, PM$_{2.5}$ and PM$_{10}$ decreased during the campaign by roughly 46%, 13% and 27%,
respectively, while NOx exhibited only a small decrease. This shows that without the impact of long-range transport,
emission reduction measures carried out by local and surrounding cities play a significant role in defining the air
quality in Jiaxing.
Table 3 Concentrations of major pollutants under similar meteorological conditions before and during the campaign

| Period | Time | Wind speed m/s | Wind direction ° | Relative humidity % | Temperature °C | Pressure hPa | Visibility km | SO$_2$ µg/m$^3$ | NO$_2$ µg/m$^3$ | PM$_{10}$ µg/m$^3$ | PM$_{2.5}$ µg/m$^3$ |
|---|---|---|---|---|---|---|---|---|---|---|---|
| Before the | 12.2 12:00- | 3.1 | 268.0 | 59.2 | 8.2 | 102.6 | 22.8 | 39.1 | 44.4 | 89.5 | 49.4 |



| campaign | 12.4 20:00 | | | | | | | | | |
|---|---|---|---|---|---|---|---|---|---|---|
| During the campaign | 12.16 9:00-12.18 5:00 | 3.4 | 247.5 | 53.0 | 2.6 | 103.2 | 32.1 | 22.4 | 39.3 | 65.3 | 42.8 |

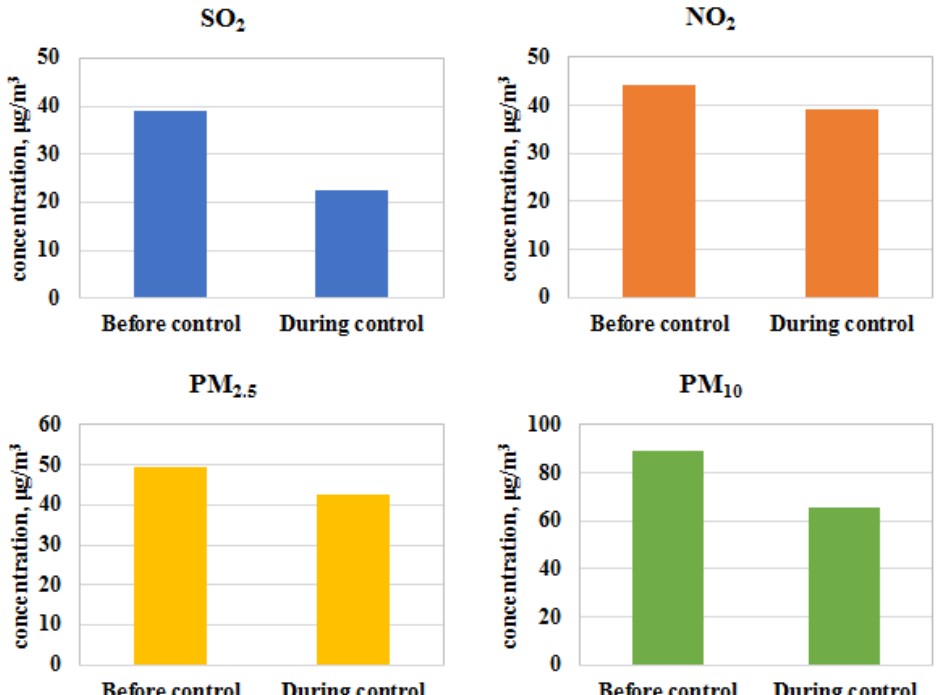

Fig. 10 Comparison between concentrations of major air pollutants in Jiaxing before and after the campaign under same meteorological conditions

There were two regional pollution episodes that occurred during the campaign. The first was on December 10-12 caused by the southward motion of northern weak cold air. Polluted air masses from south-eastern Shandong peninsula passed through central eastern Jiangsu and into northern Zhejiang, affecting the air quality in Jiaxing. During this period, the average daily PM$_{2.5}$ concentration in Jiaxing was 145.7 µg/m$^3$, higher than the regional average, and its major chemical components were nitrate (31%), sulphate (18%) ammonium (13%) and organic carbon (13%), with obvious regional secondary pollution characteristics.





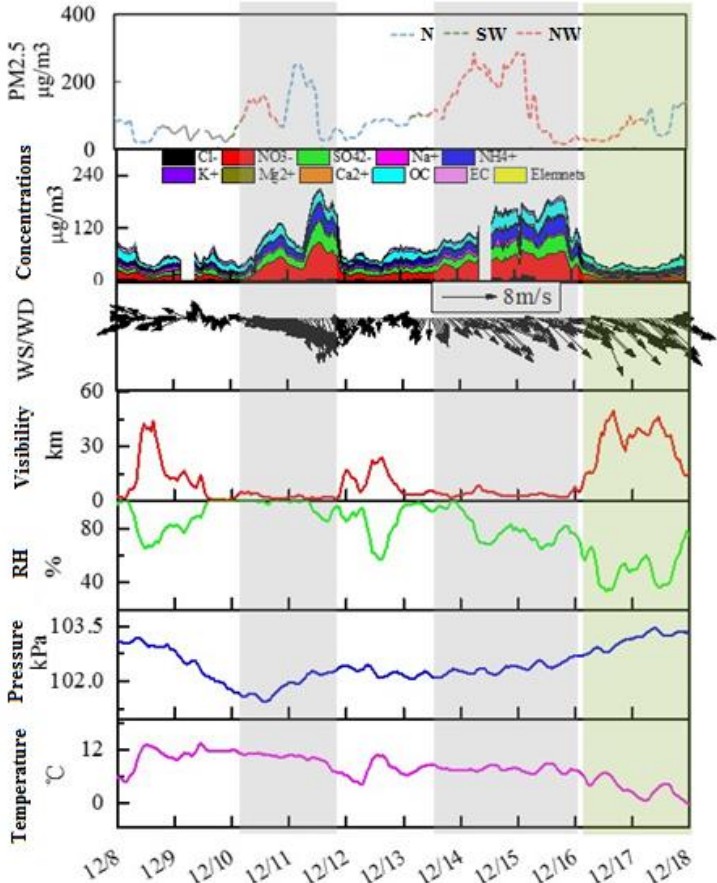


Fig.11 Changes in air quality and meteorological parameters in Jiaxing City during the campaign

The second episode occurred from December 14-15, and was caused by the transit of northwesly strong cold
air. Polluted air masses came from the northwest direction, moved rapidly to the southeast, passed through Shanxi,
Hebei, west Shandong, east Anhui and west Jiangsu and ultimately into Zhejiang province. The air masses left
China through south-eastern Zhejiang on the early morning of the 16th. The YRD region was strongly affected by
the transport of the polluted air mass, with heavy pollution appearing and lasting for about one day over the YRD
region from north to south. $PM_{2.5}$ peaked in Jiaxing on the 15th with a daily average of 201.6 µg/m$^3$. The main
chemical components of $PM_{2.5}$ during the episode were nitrate (25%), sulphate (14%), ammonium (12%) and
organic carbon (13%), which is consistent with an aged air mass as well as regional secondary pollution
characteristics.
The regional linkage was initiated from December 16 to December 18, combined with favourable mixing
conditions brought by the cold front. The overall air quality in the YRD region during this time period was good,
with an average daily $PM_{2.5}$ concentration in Jiaxing of 45 µg/m$^3$. The major chemical components during this





cleaner period were organic carbon (26%), nitrate (16%), ammonium (12%) and sulphate (9%), with some newly
formed particles and no obvious regional pollution characteristics, suggesting that air pollutants were mainly derived
from local emissions.

**397    3.3 Emission reductions during the campaign**

**398    3.3.1 Control Measures adopted to reduce air pollutant emissions**

The air quality assurance campaign for the 2nd World Internet Conference was from December 8 to December

18. In order to ensure the air quality during the conference, three provinces and Shanghai municipality in the YRD
region carried out joint control measures and established key areas, strict control areas, control areas and extension
areas, respectively, according to the degree of pollution impact from each region on the air quality in Jiaxing. Among
them, key areas and strict control areas included Zhejiang province (including Hangzhou, Ningbo, Huzhou, Jiaxing
and Shaoxing), Shanghai (including Jinshan and Fengxian), Jiangsu province (including Suzhou and Wuxi) and
Anhui province (including Xuancheng, Ma'anshan and Wuhu), as shown in figure 12.

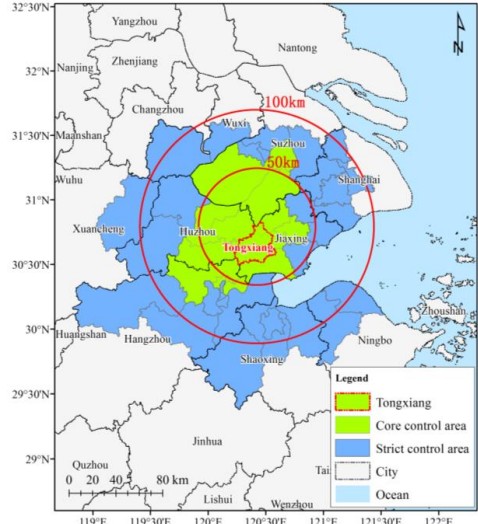

Fig.12 Controlled regions in the Action Plan for Air Quality Control during the World Internet Conference

The following measures were taken in key areas and control areas: (1) Strictly control emissions from coal-

burning power plants: reduce emissions from power plants which have not completed ultra-low emission
transmission processes by 50% in key areas and by 30% in control areas. (2) Reduce emissions from key enterprises:
production restriction or suspension would be imposed on industries including cement, steel, construction materials,
petrochemicals, chemicals, casting, leather, non-ferrous metals, plate glass, pharmaceuticals, surface spraying and
printing. All key enterprises in key areas were discontinued (maximum production limits will be imposed on steel
and petrochemical industries), while key enterprises in control areas cut emissions by 30%. Enterprises which could





not meet the emission standards in a stable way, do not have facilities for exhaust gas treatment or cannot operate
facilities normally were to be discontinued. Petrochemical and chemical enterprises were forbidden to turn on/off
for operation or maintenance. (3) Strictly control motor vehicle pollution: in the core areas in Zhejiang province,
motor vehicle restrictions were implemented, which means that low-speed trucks were forbidden to pass except for
people's livelihood-related activities. Vehicles which had not obtained valid qualifications for environmental
inspection were prohibited on the road. (4) Control dust pollution: Controlled work sites were suspended in key
areas and control areas. Dust materials were forbidden to be transported within key neighbourhoods. Dust control
measures were implemented on renovation operations at ports, docks, railway stations and commercial concrete
mixing stations and on materials storage yards. (5) Control other sources of pollution: in key areas and control areas,
oil storage facilities, gas stations or tank trucks which were not equipped with facilities for recovery of oil and gas
or facilities that could not operate normally were forbidden to sell or transport oil products. Open air barbecue,
garbage burning or straw burning in the open air were prohibited. All the primary schools, secondary schools,
kindergartens, institutions and public institutions in Jiaxing were given a three-day vacation.
Zhejiang province initiated control measures on December 8, which included: First, strictly control emissions
from coal-burning power plants through the use of low-sulphur coal and production restrictions; Second, cut
emissions from key enterprises through measures such as production restriction and suspension; Third, all
enterprises that cannot meet the emission standards in a stable way shall be discontinued; Fourth, rubbish and straw
burning shall be strictly supervised. During the period, 109 coal-burning power plants in total had imposed
production restrictions in Hangzhou, Ningbo, Huzhou, Jiaxing, Shaoxing and Jinhua. A total of 1331 key enterprises
reduced their operations among which 720 were restricted from production. Similarly, 3950 key controlled
construction sites were suspended from operation, 83 concrete enterprises were restricted from production, 444 sites
of straw burning were regulated, and 296 "black chimneys" were inspected.
As a neighbouring city, the Shanghai municipality released their Action Plan for Air Quality Control at the
World Internet Conference in 2015, focusing on key areas such as Jinshan district, Fengxian district, Shanghai
chemical industry and Shanghai petrochemical industry, and implemented emissions controls on petrochemical,
steel, chemical, coating and printing industries. On the morning of December 14, temporary control measures were
initiated. In total, 313 key enterprises and over 1000 key buildings and municipal sites were under the control of the
city, more than 320 construction sites were suspended from operation, and over 600 yard terminals strengthened
dust control measures. The ban on Yellow Label cars and the restriction on diesel vehicles within the middle ring
in Shanghai were implemented. On the morning of December 15, the Yellow Alert emergency plan was launched.
On that day, 643 key enterprises were controlled, among which 178 were restricted from production. In total, 174





construction sites, 310 demolition sites, 66 municipal road sites and 44 dock yards were suspended from operation,
along with an increase in rinsing and cleaning frequency for 2060 roads to reduce fugitive dust emissions.
Jiangsu province expanded key areas for air quality control from two districts in Suzhou (Wujiang and
Wuzhong) to three cities (Suzhou, Wuxi and Changzhou). In total, 8 power plants adopted high-quality coals in
Suzhou, another 8 power plants limited their production by 30%, and over 60 key enterprises took measures such
as restricting the production and shutting down of coal-fired boilers. Construction work sites in control areas were
suspended from operation. In Wuxi, thermal power enterprises adopted high-quality low-sulphur coals, electricity
power enterprises and 82 key enterprises were restricted or suspended from production.
In Anhui province, three control areas (Ma'anshan, Xuancheng and Wuhu) and 6 extension areas including
Anhui all developed and implemented control programs. During the campaign, 23 coal-fired power plants were
controlled for low emission, 126 enterprises were restricted from production and 287 construction work sites were
controlled. Among which, 3 cement enterprises in Xuancheng limited their production by 30%, another 3 cement
enterprises and 2 chemical enterprises limited their production by 50%, 9 construction sites were suspended from
operation, lime production and process enterprises were discontinued, quarrying and stone transportation were
prohibited, and other non-coal mines were discontinued.
During the campaign, the YRD region was frequently affected by unfavourable weather conditions such as
pollution transportation from the north. There were four distinct meteorological regimes, which occurred in Jiaxing
and its surrounding cities. Therefore, some cities further took stricter pollution reduction measures. Shanghai
municipality started temporary control of heavy pollution on December 14 and initiated the yellow warning for
heavily polluted weather on December 15. On that day, 643 key enterprises were controlled, 178 enterprises were
restricted from production, over 1000 dusty construction sites were discontinued and the rinsing and cleaning
frequency increased for over 2000 roads. Starting from December 11 in Jiangsu province, 1260 enterprises were
restricted from production, 1429 enterprises were discontinued, and all construction sites in control areas were
suspended from operation. Emergency control measures were initiated on December 15, strengthening control
efforts for industries, work sites and motor vehicles.
**3.3.2 Emissions reduction estimation**
Based on the implementation of control measures in all areas during the conference and whether each area had
effectively implemented control measures on December 8-18, regional emission reductions have been assessed. It
is estimated that emission reductions of $SO_2$, $NO_x$, $PM_{2.5}$ and VOCs caused by production restriction in regional
industrial enterprises are 2867.8 tons, 3064.7 tons, 2165.5 tons and 5055.4 tons, respectively. Emission reductions





of various pollutants caused by the restrictions on motor vehicle traffic are estimated as 4.7 tons of $SO_2$, 326.9 tons
of NOx, 36.1 tons of $PM_{2.5}$ and 452.5 tons of VOCs. Emission reduction of $PM_{2.5}$ caused by dust control was
estimated as 266.0 tons. Therefore, it can be seen that emission reductions mainly come from industrial sources,
while motor vehicle restrictions contributed greatly to emission reductions of NOx and VOCs, and dust control
contributed 10% to emission reductions of $PM_{2.5}$.
When looking at specific industries, the electricity power industry contributed most to the emission reductions
of $SO_2$ and NOx at 49.7% and 46.9%, respectively, followed by the chemical industry, building materials industry,
steel industry and petrochemical industry with a total contribution from all four sectors to emission reductions of
$SO_2$ and NOx of 42.0% and 47.2%, respectively. For $PM_{2.5}$, the building materials industry contributed the most at
62.0%, followed by steel and processing industry, power industry and non-ferrous smelting and process industry
with a contribution of 14.3%, 13.1% and 8.1%, respectively. For VOCs, the emission reduction sectors are mainly
chemical, petrochemical and machinery manufacturing sectors with a total contribution of 65.7% and individual
contributions of 25.1%, 23.2% and 17.4%, respectively. In addition, metal products processing, building materials
and steel and processing sectors also contributed significantly to emission reductions of 13.4%, 8.0% and 6.5%,
respectively.
In terms of the regional distribution of emission reductions, Jiaxing, Hangzhou, Suzhou and Shaoxing have the
largest contribution of around 80%. These four cities contribute 87% to the total emission reduction of $PM_{2.5}$.
Combing all control measures, total emission reductions of $SO_2$, $NO_x$, $PM_{2.5}$ and VOCs are estimated as 2872.5
tons, 3391.6 tons, 2467.6 tons and 5507.9 tons, respectively, which accounts for 10%, 9%, 10% and 11%,
respectively, of the total urban emissions. It is worth mentioning that if we consider the emergency emission
reduction measures for heavy pollution during the campaign, the amount of emission reduction for all pollutants
and the proportion of their emission reductions would be even larger. Table 4 shows the percentage and the amount
of emission reductions for pollutants under various control measures.

Table 4 Emission reduction estimations for various control measures

| Province | City | Sector | Amount of emission reduction (tons) | | | | Percentage of reduction | | | |
|---|---|---|---|---|---|---|---|---|---|---|
| | | | $SO_2$ | NOx | $PM_{2.5}$ | VOCs | $SO_2$ | NOx | $PM_{2.5}$ | VOCs |
| Zhejiang | Jiaxing | Industries and enterprises | 925.6 | 709.5 | 462.3 | 1872.7 | 56% | 58% | 64% | 80% |
| | Huzhou | | 414.8 | 585.6 | 602.5 | 514.0 | 46% | 37% | 47% | 53% |
| | Hangzhou | | 657.2 | 654.1 | 476.2 | 1043.2 | 36% | 42% | 59% | 33% |
| | Ningbo | | 59.1 | 65.3 | 107.5 | 84.0 | 32% | 30% | 37% | 33% |
| | Shaoxing | | 365.9 | 414.8 | 403.9 | 678.7 | 34% | 38% | 62% | 31% |
| Shanghai | Shanghai | | 253.6 | 368.7 | 83.6 | 796.1 | 9% | 7% | 6% | 8% |
| Jiangsu | Suzhou | | 89.4 | 34.9 | 10.2 | 11.4 | 3% | 1% | 1% | 1% |





| Province | City | | | | | | | | | |
|---|---|---|---|---|---|---|---|---|---|---|
| Anhui | Wuxi | | 94.4 | 163.0 | 10.2 | 55.3 | 12% | 10% | 1% | 5% |
| | Xuancheng | | 7.8 | 68.8 | 9.1 | 0.0 | 15% | 42% | 28% | 0% |
| | **Sub-total** | | **2867.8** | **3064.7** | **2165.5** | **5055.4** | **23%** | **19%** | **27%** | **19%** |
| Zhejiang | Jiaxing | Motor vehicles | 2.3 | 157.7 | 16.4 | 211.3 | 46% | 53% | 38% | 25% |
| | Huzhou | | 0.7 | 48.4 | 6.2 | 81.0 | 23% | 24% | 19% | 12% |
| | Hangzhouy | | 1.7 | 120.8 | 13.5 | 160.2 | 8% | 15% | 20% | 20% |
| | **Sub-total** | | **4.7** | **326.9** | **36.1** | **452.5** | **15%** | **25%** | **25%** | **19%** |
| Zhejiang | Jiaxing | | / | / | 119.5 | / | / | / | 100% | / |
| | Huzhou | | / | / | 11.1 | / | / | / | 10% | / |
| | Hangzhou | | / | / | 26.6 | / | / | / | 10% | / |
| | Ningbo | Dust control | / | / | 28.8 | / | / | / | 5% | / |
| | Shaoxing | | / | / | 5.8 | / | / | / | 5% | / |
| Shanghai | Shanghai | | / | / | 69.3 | / | / | / | 6% | / |
| Jiangsu | Suzhou | | / | / | 2.7 | / | / | / | 1% | / |
| | Wuxi | | / | / | 1.8 | / | / | / | 1% | / |
| Anhui | Xuancheng | | / | / | 0.4 | / | / | / | 1% | / |
| | **Sub-total** | | / | / | **266.0** | / | / | / | **9%** | / |
| | **In total** | | **2872.5** | **3391.6** | **2467.6** | **5507.9** | **10%** | **9%** | **10%** | **11%** |


**3.4 Quantitative estimates of the contribution of meteorological and control measures to air quality improvement**

503

**3.4.1 PM$_{2.5}$ concentration improvement in Jiaxing**


The WRF-CMAQ air quality model, combined with observations, was used to evaluate the reduction in PM$_{2.5}$
concentration in Jiaxing due to the emission reductions achieved through the campaign. This analysis utilized two
model simulations to assess the impact of the emission reductions: 1) a baseline scenario, which utilized an
uncontrolled emission inventory (i.e., the emissions that would have occurred without the campaign), and 2) an
emission inventory, which reflects the emission reductions achieved by the campaign. Figure 13 shows the time
series of PM$_{2.5}$ observed concentrations and the percent change in PM$_{2.5}$ after the air quality control measures were
implemented. It can be seen that the reduction in PM$_{2.5}$ concentrations in Jiaxing varies with time. The reduction in
PM$_{2.5}$ was the most significant on December 8-9 with a maximum reduction of 56%. The percent reduction in hourly
PM$_{2.5}$ during the conference (December 16-18) ranged between 2%-24%, while the average decrease in PM$_{2.5}$
concentration was 5.8μg/m$^3$ with an average improvement of about 12.9%. During the campaign from December 8
to December 18, average PM$_{2.5}$ concentrations decreased by 10.5 μg/m$^3$ with an average decrease of 14.4%.
However, Although there are many control strategies implemented, the effects during 12/14-12/16 are low. As
described in section 3.1.2, the prevailing wind direction during this period is NW, and Jiaxing experienced a heavy
pollution process with the transit and transportation of strong cold air. Therefore, we can not see obvious effect
without strong upwind precursor emissions reductions.



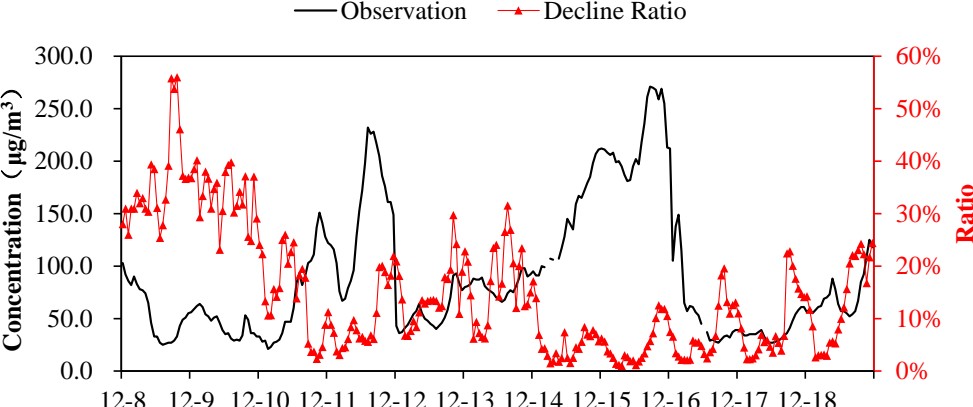


Fig. 13 Time series of observed PM2.5 and the percent reduction resulting from the implementation of air quality control measures
Figure 14 shows the reduction in daily average PM2.5 concentrations in Jiaxing resulting from the emission
reductions associated with the Action Plan for Air Quality Control at the World Internet Conference. As can be seen
from the figure, the improvement in PM2.5 before the conference (December 8 and 9) was relatively significant,
with a daily average decline of roughly 31% and 35%, respectively, which corresponds to a decrease of around 17
μg/m³. The reduction in PM2.5 on December 14-15, two of the days with some of the highest observed PM2.5, was
relatively low at around 6%, while daily average PM2.5 concentrations on those days decreased by around 10.0
μg/m³. The magnitude of emission reductions during those two time periods was basically the same, so it's likely
that the observed difference in PM2.5 levels was the result of meteorological differences, and in particular, enhanced
transport of polluted air into Jiaxing from December 14 to 15. Overall, under the influence of regional control
measures for emission reductions from December 8 to December 18, PM2.5 daily average concentration decreased
by 5.5%-34.8% with an average of 14.6% or 10 μg/m³.

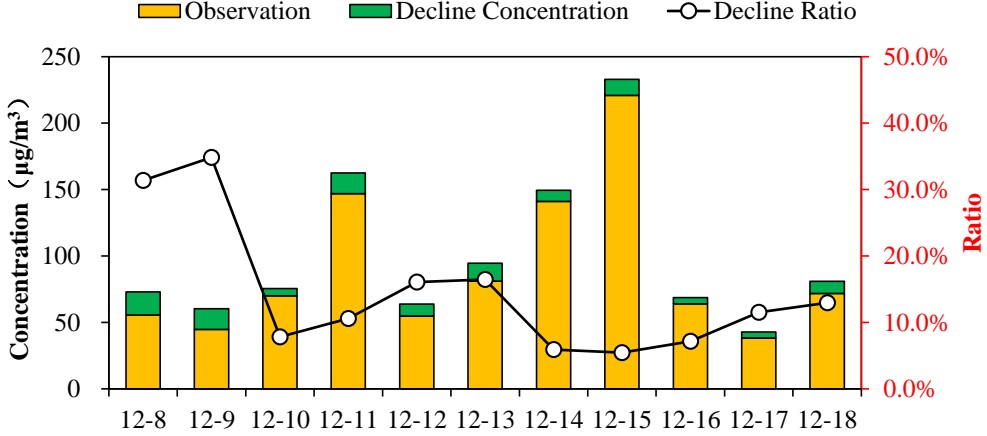

Fig.14 Percent reduction in PM2.5 resulting from the control measures





### 3.4.2 PM$_{2.5}$ concentration improvement across regions

Figure 15 shows the spatial distribution of PM$_{2.5}$ concentrations in the Yangtze River Delta region from December 8 to December 18 in the baseline scenario and the campaign scenario. As can be seen from the figure, southern Jiangsu, Shanghai and northern Zhejiang in the central YRD region had relatively high PM$_{2.5}$ concentrations, which is consistent with the typically more serious pollution levels in autumn and winter in the YRD region. Under the influence of regional control measures, PM$_{2.5}$ average concentrations declined significantly in Jiaxing, Hangzhou and Huzhou, especially at the junction of these three cities, with a slight improvement in central southern Zhejiang too. The average percent reduction in PM$_{2.5}$ concentrations in Jiaxing, Hangzhou and Huzhou was about 6%-20%. Meanwhile, given that the prevailing winds are north-westerly in winter, there was also some improvement in central and southern Zhejiang.

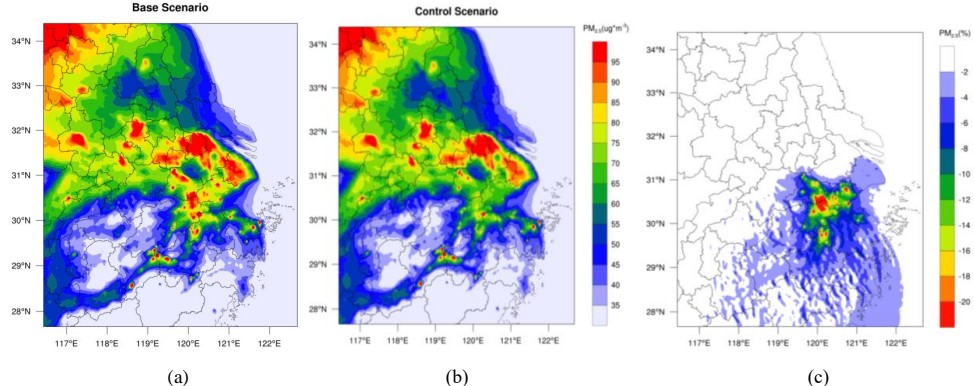

(a)                                    (b)                                    (c)

Fig. 15 Spatial distribution of PM$_{2.5}$ concentrations in the Yangtze River Delta region under the baseline scenario (a) and the campaign scenario (b), and the percent reduction in PM$_{2.5}$ throughout the YRD region (c)

### 3.4.3 Regional contributions of PM$_{2.5}$ concentration improvement in Jiaxing

Figure 16(a) shows the percent reduction in PM$_{2.5}$ daily average concentrations from December 13 to December 18 after control measures were implemented in Jiaxing and regionally. The reduction in PM$_{2.5}$ was the results of both local controls, as well as regional controls which reduced pollution in the air masses transported into Jiaxing. Overall, modelling suggests that the regional controls reduced PM$_{2.5}$ levels in Jiaxing between 5.5%-16.5% (9.9% average), while local control measures contributed 4.5%-14.4%, with an average of 8.8%.

Figure 16(b) shows the average contribution of local emissions reductions in Jiaxing and in the YRD region over the entire campaign (Dec.13-18), as well as the corresponding improvement in PM$_{2.5}$ levels in Jiaxing. During this period, PM$_{2.5}$ daily average concentration declined by 4-13 μg/m$^3$, while there were differences in the contribution of regional remission reductions and local emission reductions in Jiaxing during different periods. Overall, local control measure in Jiaxing had the largest impact on PM$_{2.5}$ levels and accounted for 89% of the decline in PM$_{2.5}$, while regional control measures contributed the remaining 11%.



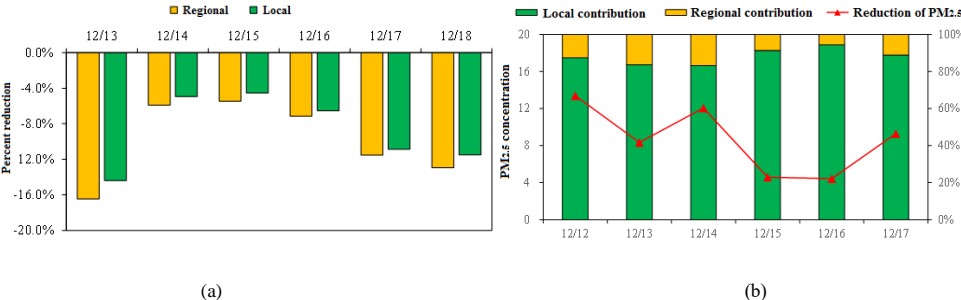

(a)                                                          (b)

Fig. 16 Percent reduction in daily average PM2.5 concentrations from December 13 to December 18 after implementation of the
control measures across the region and in Jiaxing (a) and Contribution of local and regional emissions reductions in Jiaxing, and the
resulting improvement of daily average PM2.5 concentrations in Jiaxing (b)

**3.5 Optimisation scenario analysis of regional linkage control measures**

**3.5.1 Optimization scenario settings**

In order to further analyse the optimisation potential of air quality control measures for major events and enhance the effectiveness of the control measure scheme design, three control measure optimisation scenarios have been set on the basis of the evaluation scenario (Base) after the implementation of air quality control measures during the conference. These scenarios include local emission reductions in Jiaxing under stagnant meteorological conditions, where local emission accumulation is the main contributor to the pollution process (Sce.1), and the emission reduction scenario where transport of polluted air masses into Jiaxing is a major contributor to the PM$_{2.5}$ levels in Jiaxing. In order to investigate the transport processes further, the latter scenario was further divided into a scenario 24 hours in advance (Sce.2) and a scenario 48 hours in advance (Sce.3). Table 5 describes the details of each scenario.

Table 5 Control measure optimization scenario settings

| Scenario name | Scenario settings | Emission reduction regions | Emission reduction measures | Starting time |
|---|---|---|---|---|
| Base | Regional emission reduction | All the cities and areas involved in the campaign scheme | All control measures mentioned in the campaign scheme | December 8 |
| Sce.1 | Local emission reduction in Jiaxing | Jiaxing | Control measures in Jiaxing mentioned in the campaign scheme | December 8 |
| Sce.2 | Emission reduction through transportation channels 24 hours in advance | Cities located in the northwest transportation channel of Jiaxing | Cut down industrial sources by 30% | December 13 |
| Sce.3 | Emission reduction through transportation channels 48 hours in advance | Cities located in the northwest transportation channel of Jiaxing | Cut down industrial sources by 30% | December 12 |

Figure 17 shows the cities that primarily influence the polluted air masses transported into Jiaxing, where the





transport channels were determined through backward trajectory analysis. These cities include Huzhou in Zhejiang
province, Suzhou, Wuxi, Changzhou, Nanjing, Zhenjiang, Huai'an, Suqian and Suzhou in Jiangsu province and
Suzhou, Huaibei, Bozhou, Bengbu, Chuzhou and Ma'anshan in Anhui province. Each of these cities took measures
to reduce emissions by limiting production from industry industries by 30%.

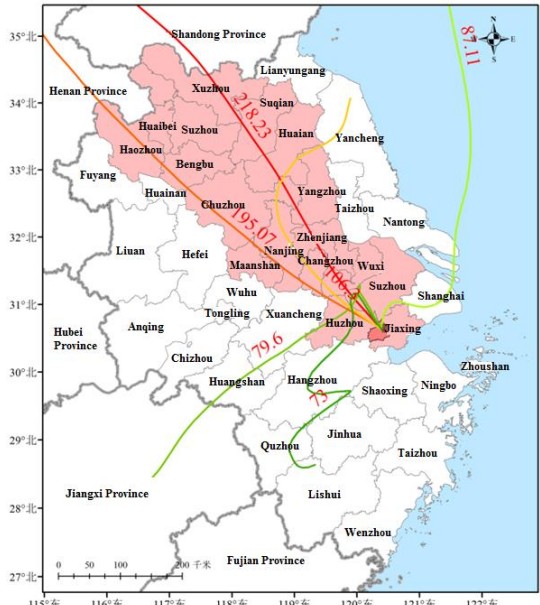

Fig. 17 Cities involved in the transportation channel and the emission reduction channel

The WRF-CMAQ modelling system was used to analyse and compare the air quality improvement effect under

different pollution process in four scenarios.
**3.5.2 Analysis of optimization scenario effects**

In order to evaluate the effect of the different starting time for the same control measures, and the same starting

time for local and regional control measures, we conducted four scenarios. Figure 18 shows the percent reduction
in daily average $PM_{2.5}$ concentrations in Jiaxing City from December 13 to December 18 under the regional emission
reduction scenario, the Jiaxing local emission reduction scenario and the transportation channel emission reduction
scenario. Overall, there are differences in the distribution of $PM_{2.5}$ under the different scenarios. The air quality
improvement due to the regional emission reductions was higher than that of local emission reductions in Jiaxing,
and lower than that of channel emission reductions.



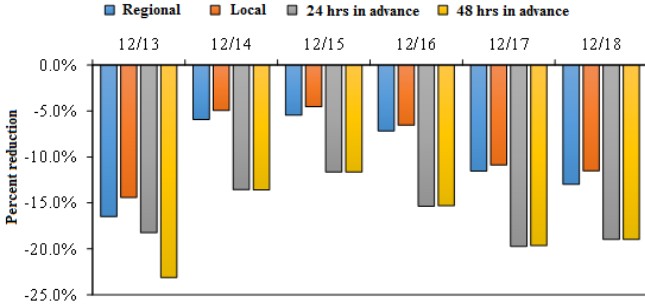


Fig. 18 Decline rates of PM$_{2.5}$ daily average concentrations in Jiaxing under different scenarios
(1)**Effect of local emission reductions in Jiaxing**
By comparing the effect of local emission reductions in Jiaxing (Sce.1) and the effect of regional emission
reductions (Base), we can see that PM$_{2.5}$ daily average concentrations in Jiaxing declined by around 5.5%-16.5%
under the regional emission reduction plan (regional emission plan including the local emissions control) from
December 13 to December 18 and by around 4.5%-14.4% under the local emission reduction plan. Local emission
reductions in Jiaxing contributed 83%-94% to the emission reduction effect. Therefore, local emission reduction in
Jiaxing is the key factor in improving the local air quality.
Compared with the channel emission reduction scenario 24 hours in advance (11.6%-18.2%), local emission
reductions also contributed more than 50% to the improvement effect on December 13, 17 and 18. Therefore, local
emission reductions contributed most to the air quality improvement effect in Jiaxing, indicating that local areas are
still the most important control areas during the campaign.
(2)**Effect of emission reductions through transportation channels**
As mentioned above, during the large-scale transport of heavily polluted air masses into the Yangtze River
Delta region from December 14 to December 15, the PM$_{2.5}$ pollution in Jiaxing was significantly affected. Under
the local emission reduction scenario (Sce.1) and the regional linkage emission reduction scenario (Base), PM$_{2.5}$
daily average concentrations in Jiaxing decline by only 4.5%-5.9%. If a 30% reduction in emissions from industrial
sources in the upwind transportation channel is implemented, PM$_{2.5}$ daily average concentrations in Jiaxing declined
by 11.6%-13.6%, while local emission reductions contributed less than 40% to the improvement of PM$_{2.5}$. Therefore,
to reduce PM$_{2.5}$ under these large-scale transport conditions, in addition to intensifying local emission reduction
efforts, it is more effective to prevent and control such pollution by adopting emission reductions of industrial
sources over key transportation channels, especially for elevated sources.
In this study, the main transportation channel involved is the northwest transportation channel in control areas,
which basically represents the typical winter transportation channel in the region. A well-designed management





plan for the main transportation channel is necessary to ensure the air quality in autumn and winter is improved, in
addition to reducing local emissions.
(3)**Effect of the starting time for channel emission reductions**
According to the comparisons between the emission reduction scenario 24 hours in advance (Sce.2) and the
emission reduction scenario 48 hours in advance (Sce.3) during the large-scale $PM_{2.5}$ transport, we can see that if
we take December 13 as the target and adopt channel emission reductions 48 hours in advance, $PM_{2.5}$ daily average
concentrations will decline by 23.1% when compared to the baseline scenario, which is significantly better than the
improvement achieved by the emission reduction scenario 24 hours in advance (18.2%). Therefore, early measures
to reduce emissions will lead to the improvement of air quality.
If we focus on the conference period (December 16-18), $PM_{2.5}$ daily average concentrations will both decline
by 15.3%-19.7% under the two channel emission reduction scenarios, indicating a close improvement effect.
Therefore, during the pollution process when local emissions are the main contributor, local emission reductions
should be the top priority with no difference between channel reductions 24 hours in advance and 48 hours in
advance. If transportation emissions are the main contributor to the pollution, adopting channel reductions 48 hours
in advance can bring about more improvement effect than 24 hours in advance.
**3.6 Comparisons with other air quality guarantee events**
Besides the 2nd World Internet Conference, China has hosted many other mega events in recent years, like
2008 Beijing Olympics, 2010 Guangzhou Asian Games, 2014 Beijing APEC, 2014 Nanjing Youth Olympics and
2015 Victory Parade. To guarantee the better air quality and protect people's health, local government had
implemented numerous short-term stringent strategies to guarantee air quality during these events, which
significantly reduced the emissions and concentrations of air pollutants in these cities and surrounding area. Through
banning yellow-label vehicles from driving, stopping all construction activities and requiring heavily polluting
factories to reduce their operating capacities or completely shut down during the 2008 Olympics, the mean
concentrations of $SO_2$, $PM_{2.5}$ and $NO_2$ in Beijing and its surrounding area were reduced by 51.0%, 43.7% and 13%
compared to the period before Olympics, and concentrations of $O_3$, $SO_2$, CO and NOx decreased 23%, 61%, 25%
and 21% compared to previous years (Wang, et al., 2009; Wang, et al., 2010). During the 2010 Asian Games, the
Guangzhou government made great efforts to improve the air quality, such as controlling emissions from industries
and transportation restrictions, requiring vehicles to drive only on alternate days depending on license plate numbers,
and prohibiting all construction activities. The emissions of $SO_2$, $NO_x$, $PM_{10}$, $PM_{2.5}$ and VOCs were reduced by
41.1%, 41.9%, 26.5%, 25.8% and 39.7%, respectively, leading to a dramatic decrease on concentrations of SO2,



NO2, PM10 and PM2.5 in Guangzhou (Liu, et al., 2013). During the 2014 Asia-Pacific Economic Cooperation
(APEC) period, the average concentrations of PM2.5, PM10, SO2 and NO2 decreased by 47%, 36%, 62% and 41%
respectively by controlling emissions from traffic, industry, construction sites and so on (Tang, et al., 2015; Li, et
al., 2016; Wang, et al.,2016; Sun, et al., 2016; Wang, et al., 2015). During 2014 Nanjing Youth Olympics and 2015
Victory Parade, the local governments also successfully improved air quality by carrying out many air pollution
control measures including relocating some heavily polluting enterprises, encouraging natural gas instead of coal-
fired boilers and domestic stoves, limiting the use of cars and so on (Chen, et al., 2017; Han, et al., 2016). Also, 22
surrounding cities in the YRD region were asked to cooperate with Nanjing to close industries with high pollution
emissions. Some research papers demonstrate that, the mean concentrations of $PM_{2.5}$, $PM_{10}$, $SO_2$, $NO_2$, CO and $O_3$
in Nanjing during the Youth Olympics decreased by 35.92%, 36.75%, 20.40%, 15.05%, 8.54% and 47.15%,
respectively, compared with the average levels in July 2014 (Qi, et al., 2016). The emission reductions for SO2,
NOx, PM10, PM2.5, VOCs during the Victory Parade were 36.5%, 49.9%, 50.3%, 49.0% and 32.4%, respectively.
These results show that stringent emission reduction strategies had greatly relief the air pollution of these cities
including their surrounding area, and further improve their air quality through regional joint control measures.
Table 6 Summary of control strategies taken during different periods

| Periods | Control area | Control policies | Main achievements or targets |
|---|---|---|---|
| Beijing Olympics, 2008 | Jing-jin-ji area Inner Mongolia, Shanxi, Shandong | Yellow-label vehicles were banned from driving; personal vehicles were taken off the roads through the alternative day-driving scheme; all construction activities were halted; power plants were asked to use cleaner fuels and reduce their emissions by 30% from their levels; heavily polluting factories were ordered to reduce operating capacities or completely shut downs. | The emissions of $SO_2$, NOx, CO, VOCs, and $PM_{10}$ were reduced by 14%, 38%, 47%, 30% and 20% respectively; the concentrations of fine and coarse particulate matter were reduced by 35–43%. |
| Beijing APEC, 2014 | Jing-jin-ji area Shanxi, Inner Mongolia, Shandong | Reducing or stopping production in factories, halting production on construction sites, imposing the odd-even traffic rule for vehicles, and strengthening road cleaning measures. | Average concentrations of $SO_2$, $NO_2$, $PM_{10}$ and $PM_{2.5}$ decreased by 62%, 41%, 36%, and 47%, respectively. |
| Nanjing Youth Olympics, 2014 | Nanjing | 2630 construction sites were halted; heavy-industry factories were required to reduce manufacturing by 20%; high-emission vehicles were not allowed to drive on the road; open space barbecue restaurants were closed; over 900 electric buses and 500 electric taxis have been put into operation. | The mean concentrations of $PM_{2.5}$, $PM_{10}$, $SO_2$, $NO_2$, CO and $O_3$ decreased by 35.92%, 36.75%, 20.40%, 15.05%, 8.54% and 47.15%. |
| Beijing | Jing-jin-ji | Over 111 businesses with high emissions | The emission reductions were 36.5% |





| Victory Parade, 2015 | area, Shanxi, Shandong, Inner Mongolia, Henan | were required to stop or limit their production; the odd-even rule was implemented to restrict traffic emissions; 80% of government vehicles were prohibited on roads; trucks transporting mud and stone as well as heavy-emission vehicles were prohibited from roads; over 10000 enterprises were closed or ordered to limit production, and about 9000 construction sites were shut down. | for $SO_2$, 49.9% for $NO_x$, 50.3% for $PM_{10}$, 49.0% for $PM_{2.5}$ and 32.4% for VOCs in Beijing. |

## 4 Conclusions

**(1) The effect of restricting production in industrial enterprises is remarkable.** The power industry and related industrial enterprises in Jiaxing cut down SO2 and NOx emissions by over 50%, while the building materials industry, smelting industry and other industrial enterprises cut down PM2.5 emissions by 63%, contributing greatly to the reduction of primary PM2.5 concentrations. The petrochemical industry, chemical industry and other related industrial enterprises cut down VOCs emission by 66% in total, contributing greatly to the reduction of PM2.5 formed through the conversion of precursor species. The observation data of PM2.5 components suggest that the relative contribution of secondary components dropped significantly during the conference. Production restriction or suspension for industrial enterprises is the main contributor to emission reductions for various pollutants during the campaign, which resulted in the largest improvement in air quality.

**(2) Motor vehicle pollutant emissions declined significantly.** In Jiaxing, motor vehicle restrictions were fully implemented during heavy pollution days, temporary traffic control was implemented during certain periods, and enterprises and institutions had a three-day vacation during the conference. Emission reduction rates for various pollutants from motor vehicle emissions were around 40%-50%. Motor vehicle emission reduction measures contributed to the total emission reductions of nitrogen oxides by 18.2%, fine particles by 3.4% and volatile organic compounds by 10.1%.

**(3) The effect of dust control measures is remarkable.** During the conference, all the work sites in Jiaxing and 3950 work sites in total in Zhejiang province were suspended from operation. Measures of increasing frequency for road cleaning activities greatly lowered the dust emissions. Speciation of the measured $PM_{2.5}$ suggest that the mass concentration of crust material, which is greatly affected by dust, decreased by 14% compared to measurements after the conference, indicating the effectiveness of dust control measures.

**(4) Regional linkage between surrounding areas played an important role.** $PM_{2.5}$ is a typical regional air pollutant, with obvious regional transportation characteristics. In accordance with the requirements of the campaign scheme, eight cities around Jiaxing have actively implemented emissions reduction measures. During the campaign,



PM2.5 concentrations in eight surrounding cities and south-eastern Zhejiang also declined with obvious regional
synergies.
It is worth noting that the implementation of control measures has also had a negative impact on the economy
and the society in the short term while improving the air quality. For example, production restriction or suspension
on a large number industrial enterprises were taken at great economic costs, and motor vehicle restriction had a
large impact on the society.
**(5) Suggestions on emission reduction plans:** Local emission reductions shall be supplemented by regional
linkage. Assessment results show that local emission reductions play a key role in ensuring air quality. Therefore,
it is recommended that a synergistic emission reduction plan between adjacent areas with local pollution emission
reductions as the core part should be established and strengthened, and emission reduction plans for different types
of pollution through a stronger regional linkage should be reserved. Strengthen the pollution reduction in the upper
reaches along the transportation channel. It is especially crucial to enhance pollution emission reductions in the
upper reaches of the channel since long-distance transport of pollution is a problem. This is especially true for key
industrial sources and elevated sources. Considering that polluted air mass transportation is more frequent in winter,
it is necessary to develop emission reduction plans for different pollution transportation channels, combined with
forecasting and warning mechanisms which could be initiated on time.

**Acknowledgements.**
This study was financially supported by the "Chinese National Key Technology R&D Program" via grant No.
2014BAC22B03 and he National Natural Science Foundation of China (NO. 41875161). We also thank the Joint
pollution control office over the Yangtze River Delta region for co-ordinating the data share.

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
