# Peer review of "Evaluation on the effect of regional joint control measures in changing"

_Atmospheric Chemistry and Physics, 2018_

## Referee Comment (RC1) · Anonymous Referee #1 · 1 Feb 2019

Review of "Evaluation on the effect of regional joint control measures in changing photochemical transformation: A comprehensive study of the optimization scenario analysis" by Li et al.

This manuscript investigated the effects of joint local and regional regulations on air pollution during the 2nd World Internet Conference held in Jiaxing, Zhejiang. Both modeling and measurements were used for the evaluation. The authors performed careful case studies by controlling the meteorological conditions, air mass backtrajectory, etc. Different emission reduction plans were proposed based on different scenarios. In particular, it is recommended to implement regulation along the transport channel to the receptor-site. This is an important study to develop effective control strategies to mitigate air pollution in China. Overall, the manuscript is well-written and the analysis is solid. I recommend publication after minor revision.

Comments

1.     Line 202. Please show the equations to calculate the metrics. Also, "Index of Agreement" should be as "IOA".

2.     Figure 2. Please include NMB, NME, and IOA (Table 2) in the figure.

3.     Line 241-244. This sentence has grammatical error.

4.     Figure 3-7. In panel (d), please specific if the PM2.5 time series is from modeling or measurement.

5.     Figure 8 and Figure 9. These two figures are really intriguing. Why is "[SO2] after control" is similar to "[SO2 during control]", but "[SO4] after control" is much higher than "[SO4 during control]"? The opposite trend is observed for [NO2] and [NO3]. Please make similar figure for the [SO2]+[SO4] and [NO2]+[NO3], which should better represent the effect of regulation. Another potential plot is the partitioning of SOx and NOx (e.g., SO2/(SO2+SO4)). Interesting chemistry may be inferred from these analyses. Also, can the model reproduce these observations? Last comment, please consider to change the x-axis label from dates to "before/during/after regulation".

6.     Line 511. "reduction in PM2.5 concentrations" is not accurate. It should be "PM2.5 decline ratio".

7.     Figure 14. It is surprising to see that the decline ratio is typically ~10% after such strict regulation policies. What are the sources of the residual PM? From transport?

8.       Effect of local emission reductions in Jiaxing and Figure 18. Regional control only has slight extra benefit over local control. Does it suggest that less strict regulation should be implemented in nearby cities?

---

## Referee Comment (RC2) · Anonymous Referee #2 · 13 Feb 2019

Review for "Evaluation on the effect of regional joint control measures in changing photochemical transformation: A comprehensive study of the optimization scenario analysis"

This paper investigates the effect of regional control during the 2nd World Internet Conference from December 16 to December 18, 2015. They analyzed the meteorology condition, observed air pollutant concentration, and quantified the effect of air pollution control using numerical models. They found the local emission reduction plays an important role in air quality improvement and suggest that a 48-hr advance pollution

channel control before the event. Overall, this paper is well-organized and fits into the scope of Atmospheric Chemistry and Physics on the advance understanding of atmospheric chemistry process. I suggest this paper gets accepted with the following minor revisions.

Minor comments:

1. In the model performance section, the author mentioned about the underestimation of the simulated PM2.5 concentration compared to the observation. Where are the uncertainties possibly coming from? Knowing this uncertainty in the model, how do we interpret the results (possible uncertainty and limitation in the result)?

2. Some of the figure (Figure 3-7) contents are hard to read, for example, the values on the color bar on the panel (b) and contours on the synoptic maps (a). Moreover, the graph resolution is not consistent in these Figures, especially figure (c). What is the color scale in (c)?

3. Line 153: "GDAS" needs to be defined at its first appearance.

4. Line 201: . . . Index of Agreement (IOA). Same apply to Line 209: . . . and the IOA value of 0.67 to 0.70.

5. Line 340: " under static weather condition"

6. Figure 9: what is the unit of the measurement (%)?

7. Figure 11: WS/WD panel has similar information as the PM2.5 (top panel) regarding the wind direction. I suggest change the WS/WD panel to wind speed only and use contour lines to represent that.

8. Line 649-652: Please be consistent on the notification, such as SO2 PM2.5. This occurs in other sections of the manuscript, e.g. line 669-672.

---

## Referee Comment (RC3) · Anonymous Referee #3 · 19 Feb 2019

The emission reduction during the Second World Internet Conference provided a unique scenario to evaluate the chemical/physical processes affecting the air quality in Yangtze River Delta region. This paper estimated the emission reduction and simulated this scenario in a reasonable way. It provides some useful insights in the air quality management in this region. One thing is missing is this paper did not show how the chemistry works during the emission reduction period. Since sulfate and nitrate are both secondary, how they were formed and how they were affected? How did nitrate become more significant than sulfate with and without the control measures? The role

of dust emission was not paid enough attention in the discussion.

There is also a big room for improvement of overall writing. This paper is not presented consistently. It gives me a feeling that this paper is written by two different people. Later part was better presented than the first half.

Some detail suggestions:

1. Transport vs transportation

Better not to use 'transportation of air mass'. Transportation is for traffic related business. It's used for mobile emission. A better way is to say 'the transport of air mass' for the movement of air mass/pollutants/plumes.

2. Pollution vs pollutant

The used of a lot of 'pollution' in this paper is quite confusing. I think you refer it as either 'plumes' or 'polluted air masses'. Pollution is a status, it does not mean any subject and cannot be moved around. While the plumes or pollutants can be moved or transported. I'd strongly suggest the author to check all the wordings in this paper.

3. P3, line 69-70, 'Many studies. . .', 'Some have reported . . .'. Any references?

4. P3, Figure 12 may be better shown here in the introduction.

5. P4, line 101-102, 'online' and 'On-line'?

6. P4, line 108, 'consisting of' to 'such as/including'?

7. P4, line 110, 'data conform' to 'data quality conform'?

8. P5, line 137, 'with observation data and meteorological data included'. Did you used met observations for TrajStat? How?

9. P5, line 140, 1x1 degree is quite coarse. Why not just used WRF simulations?

10. P5, line 144, 'increase with the raise of distance' to 'increase with the distance'

11. P5, line 147, 'the number of total' to 'the total number of'

12. P6, 154, does that mean only PM2.5 > 75 were used?

13. P6, WRF model setup, did you used FDDA? Why not?

14. P6, WRF model setup, what are the grid size for each domain?

15. P6, WRF model setup, you may name the 3 domain as D01/D02/D03. It's very confusing here.

16. P10, "4 processes" are very confusing too. 4 episodes seems to be more reasonable.

17. P10, line 240-241, "For each of these processes, ...", confusing.

18. P10, line 255, 'ground humidity' to 'surface humidity'

19. P10, line 258, 'spread' to 'dispersion'

20. P11, Fig 3, add a few words for red line on (d)

21. P12, two "3.1.2" section

22. P13, line 295, 'slower' seems to mean 'reduced/lowered'

23. Fig3-7, color scale missing for (c)

24. P15, line 325, 'normal pollutants'?

25. P16, line 344, why not 'K' reduction? Which is a strong indicator of soil/mineral source

26. P17, Table3, Pressure is wrong

27. P18, line 378 and P19, line 388, the contributions from other PM2.5 components were 25% or 36% with transport and without transport impact. And the 'other components' should be EC and dust. EC is generally low, so the majority would be dust. If

that's true, dust PM2.5 would be the most important equal to or after sulfate. If the dust can be controlled, it's more than what has been achieved due to the control measures. Any idea what can be done to reduce the dust emissions?

28. P20, Line 393, One more evidence of other components is 33%

29. P20, section 3.3.1. This section can be more concise. If needed, Details can be moved into supplement materials. The focus here is the Table3.

30. P20, line 394, 'obvious regional pollution characteristics', what is it?

31. P28, line 589, 'percent reduction' to 'percentage reduction', 'conducted' to 'considered/investigated/discussed/etc'

32. P30, section 3.6 seems to be not that relevant here. It may be moved into the introduction or the supplement.

33. P32, line 682.'The effect of dust control measures is remarkable'. This conclusion comes from nowhere. It has not been discussed or showed in this paper. Better to prove it or remove it.

---

## Author Comment (AC1) · 27 Mar 2019

*Author Comments: Response to reviewers' comments*

Title: Evaluation on the effect of regional joint control measures in changing photochemical transformation: A comprehensive study of the optimization scenario analysis

Referees' comments:

Reviewer #1:

This manuscript investigated the effects of joint local and regional regulations on air pollution during the 2nd World Internet Conference held in Jiaxing, Zhejiang. Both modeling and measurements were used for the evaluation. The authors performed careful case studies by controlling the meteorological conditions, air mass back trajectory, etc. Different emission reduction plans were proposed based on different scenarios. In particular, it is recommended to implement regulation along the transport channel to the receptor-site. This is an important study to develop effective control strategies to mitigate air pollution in China. Overall, the manuscript is well-written and the analysis is solid. I recommend publication after minor revision.
Comments

1. Line 202. Please show the equations to calculate the metrics. Also, "Index of Agreement" should be as "IOA".

Revised. The equations have been added to the manuscript, as follows. The "I" has been revised to IOA.

The equations to calculate these statistical indexes are as follows:

$$NMB = \frac{\sum(P_j - O_j)}{\sum O_j} \times 100\% \tag{1}$$

$$NME = \frac{\sum|P_j - O_j|}{\sum O_j} \times 100\% \tag{2}$$

$$IOA = 1 - \frac{\sum(P_j - O_j)^2}{\sum(|P_j - \bar{O}| + |O_j - \bar{O}|)^2} \tag{3}$$

where $P_j$ and $O_j$ are predicted and observed hourly concentrations, respectively. $\bar{O}$ is the average value of observations. IOA ranges from 0 to 1, with 1 indicating perfect agreement between model and observation.

2. Figure 2. Please include NMB, NME, and IOA (Table 2) in the figure.

Revised, as follows.

[Figure]

3. Line 241-244. This sentence has grammatical error.

Original sentence: For each of these processes, this study has comprehensively in the integrated emission-measurement-modeling method considered the backward air flow trajectory, potential contribution source areas, meteorological conditions and the variation of PM$_{2.5}$ concentration to analyse the evolution of the observed air quality."

Revised sentence: For each of these processes, this study utilized the integrated emission-measurement-modeling method to analyze the evolution of air quality from several aspects, including the backward air flow trajectory, potential source contribution areas, meteorological conditions and the variation of PM$_{2.5}$ concentration.

4. Figure 3-7. In panel (d), please specific if the PM2.5 time series is from modeling or measurement.

It is from measurement. The original sentences "(d) PM$_{2.5}$ time series for selected sites during … " have been revised to "(d) Observed PM$_{2.5}$ time series for selected sites during …" in Fig.3-7 (Fig.4-8 in the revised manuscript).

5. Figure 8 and Figure 9. These two figures are really intriguing. Why is "[SO2] after control]" is similar to "[SO2 during control]", but "[SO4] after control" is much higher than "[SO4 during control]"? The opposite trend is observed for [NO2] and [NO3]. Please make similar figure for the [SO2]+[SO4] and [NO2]+[NO3], which should better represent the effect of regulation. Another potential plot is the partitioning of SOx and NOx (e.g., SO2/(SO2+SO4)). Interesting chemistry may be inferred from these analyses. Also, can the model reproduce these observations? Last comment, please consider to change the x-axis label from dates to "before/during/after

regulation".

It is a very good question and suggestion!

As is shown from the figure, the $SO_2$ concentrations after control is a little bit higher than during control (+5.9%). However, the $SO_4^{2-}$ after control is much higher than during control (25.8%). This is probably due to two reasons: firstly, $SO_2$ emissions and primary sulfate emissions increased after the control measures were stopped; secondly, increased $NO_2$ emissions could accelerate the formation of secondary sulfate (Cheng et al., 2016), which can be clearly shown from the SOR and NOR. Different trend is observed for $NO_2$ and $NO_3^-$, with the $NO_2$ concentrations after control much higher than during control (+9.4%), while the increase ratio of $NO_3^-$ (+9.45%) is the same. Sulfate originates from both primary emissions and secondary formation, but nitrate is mostly secondary formed. The NOR during and after regulation is the same and most of the N is in the gas phase ($NOx/(NOx+NO_3^-)$ is 0.87). Therefore, the increase of $NO_3^-$ is lower than $SO_4^{2-}$. The $PM_{2.5}$ concentration after control sharply rebounded 31.8%, indicating that both the emissions increased, and the secondary pollution formation is improved.

To better illustrate emissions and chemistry before, during and after control measures, we revised the previous figures and added another two indicators for partitioning of SOx /NOx, and SOR/NOR.

[Figure]

[Figure]

[Figure]

[Figure]

6. Line 511. "reduction in PM2.5 concentrations" is not accurate. It should be "PM2.5 decline ratio".

Revised.

7. Figure 14. It is surprising to see that the decline ratio is typically ~10% after such strict regulation policies. What are the sources of the residual PM? From transport?

The decline ratio changes with meteorological conditions even under the same emissions reduction situation, because meteorological conditions influence dispersion from primary emissions, regional transport and secondary formation. If we look at the decline ration of hourly concentrations, we can find that the decline ratio was the most significant on December 8-9 with a maximum reduction of 56%. The percentage reduction in hourly PM2.5 during the conference (December 16-18) ranged between 2%-24%. If we look at the PM2.5 decline ratio in daily average, we can see the improvement in PM2.5 before the conference (December 8 and 9) was relatively significant, with a daily average decline of roughly 31% and 35%, respectively, which corresponds to a decrease of around 17 $\mu g/m^3$. The reduction in PM2.5 on December 14-15, two of the days with some of the highest observed PM2.5, was relatively low at around 6%, while daily average PM2.5 concentrations on those days decreased by around 10.0 $\mu g/m^3$. The magnitude of emission reductions during those two time periods was basically the same, so it's likely that the observed difference in $PM_{2.5}$ levels was the result of meteorological differences. Overall, the residual PM may come from two reasons: (1) although stringent control measures have been implemented, there are still precursor emissions in this city, which accumulate and form secondary particles under specific meteorological conditions; (2) enhanced transport under different meteorological conditions, especially upwind emissions.

8. Effect of local emission reductions in Jiaxing and Figure 18. Regional control only has slight extra benefit over local control. Does it suggest that less strict regulation should be implemented in nearby cities?

Figure 18 shows the decline ratio of daily average PM2.5 concentrations under the regional emission reduction scenario, the Jiaxing local emission reduction scenario and the transport channel emission reduction scenario (24 hrs in advance and 48 hrs in advance). Air quality improvement due to regional emission reductions was slightly larger than that of local emission reductions in Jiaxing, and smaller than that of channel emission reductions. This suggests that emissions reduction in downwind cities does not

have much effect on Jiaxing's air quality. In contrast, emissions reduction based on predicted transport pathway in advance are more effective than local emissions reduction.

Reviewer #2:

Review for "Evaluation on the effect of regional joint control measures in changing photochemical transformation: A comprehensive study of the optimization scenario analysis"

This paper investigates the effect of regional control during the 2nd World Internet Conference from December 16 to December 18, 2015. They analyzed the meteorology condition, observed air pollutant concentration, and quantified the effect of air pollution control using numerical models. They found the local emission reduction plays an important role in air quality improvement and suggest that a 48-hr advance pollution channel control before the event. Overall, this paper is well-organized and fits into the scope of Atmospheric Chemistry and Physics on the advance understanding of atmospheric chemistry process. I suggest this paper gets accepted with the following minor revisions.

Minor comments:
1. In the model performance section, the author mentioned about the underestimation of the simulated $PM_{2.5}$ concentration compared to the observation. Where are the uncertainties possibly coming from? Knowing this uncertainty in the model, how do we interpret the results (possible uncertainty and limitation in the result)?

Although simulated $PM_{2.5}$ can well capture the air pollution situation, with IOA of 0.67 and 0.70, the predicted $PM_{2.5}$ is relatively lower than the observed data (NMB values are all negative). These underestimations may be due to three reasons. Firstly, winter underestimation of $PM_{2.5}$ (especially SOA) is a common issue with CMAQ or CAMx simulations over China (Hu et al., 2017; Li et al., 2016), which can be explained by a lack of model calculated oxidants or missing reactions (Kasibhatla et al., 1997) and the state-of-science of SOA formation pathways (Appel et al., 2008; Foley et al., 2010; Chen et al., 2017). Secondly, uncertainty still exists in the regional emissions inventory, including the basic emissions inventory and the control scenarios. Thirdly, the wind speed is slightly overestimated over the region, with NMB and NME of 28% and 33%, causing fast dispersion of air pollutants.
In view of these uncertainties, we mainly use observational data to interpret the photochemical change, while in Section 3.4, we should keep in mind that the secondary formation may probably be underestimated, causing the decline ratio lower than reactivity.
Text has been added to interpret the model performance and the predicted results in the model performance section 2.3.2 and section 3.4.1.

Added references:
Appel, K.W., Bhave, P.V., Gilliland, A.B., Sarwar, G., Roselle, S.J., 2008. Evaluation of the community multiscale air quality (CMAQ) model version 4.5: sensitivities impacting model performance; part II particulate matter. Atmos. Environ. 42, 6057e6066. http://dx.doi.org/10.1016/j.atmosenv.2008.03.036.
Chen, Q., Fu, T. M., Hu, J., Ying, Q., & Zhang, L. (2017). Modelling secondary organic aerosols in China. National Science Review, 4(6), 806-809.

Foley, K.M., Roselle, S.J., Appel, K.W., Bhave, P.V., Pleim, J.E., Otte, T.L., Mathur, R., Sarwar, G., Young, J.O., Gilliam, R.C., Nolte, C.G., Kelly, J.T., Gilliland, A.B., Bash, J.O., 2010. Incremental testing of the community multiscale air quality (CMAQ) modeling system version, 4.7. Geosci. Model Dev. 3, 205e226.

Kasibhatla, P., Chameides, W.L., Jonn, J.S., 1997. A three dimensional global model investigation of seasonal variations in the atmospheric burden of anthropogenic sulphate aerosols. J. Geophys. Res. 102, 3737e3759.

Li, J. L., ZHANG, M. G., GAO, Y., & CHEN, L. (2016). Model analysis of secondary organic aerosol over China with a regional air quality modeling system (RAMS-CMAQ). Atmospheric and Oceanic Science Letters, 9(6), 443-450.

Hu, J., Wang, P., Ying, Q., Zhang, H., Chen, J., Ge, X., ... & Zhao, Y. (2017). Modeling biogenic and anthropogenic secondary organic aerosol in China. Atmospheric Chemistry and Physics, 17(1), 77-92.

2. Some of the figure (Figure 3-7) contents are hard to read, for example, the values on the color bar on the panel (b) and contours on the synoptic maps (a). Moreover, the graph resolution is not consistent in these Figures, especially figure (c). What is the color scale in (c)?

We updated the precision and enlarged these figures, so that they are more clearly to read. We also added the color scale to figures (c). See revision in the revised manuscript.

3. Line 153: "GDAS" needs to be defined at its first appearance.

"GDAS" has been revised to "Global Data Assimilation System (GDAS)".

4. Line 201: : : : Index of Agreement (IOA). Same apply to Line 209:and the IOA value of 0.67 to 0.70.

Revised.

5. Line 340: " under static weather condition"

Revised.

6. Figure 9: what is the unit of the measurement (%)?

The unit is "$\mu g/m^3$", revised.

7. Figure 11: WS/WD panel has similar information as the PM2.5 (top panel) regarding the wind direction. I suggest change the WS/WD panel to wind speed only and use contour lines to represent that.

The top panel with different colors mainly indicates the trajectories at the 500m height, which can show the transport; while WS/WD means the surface wind, which can give us information regarding pollution dispersion or accumulation. Therefore, we suppose it is better to keep both.

8. Line 649-652: Please be consistent on the notification, such as SO2 PM2.5. This occurs in other sections of the manuscript, e.g. line 669-672. 2019.

We have gone through the manuscript and made edits accordingly.

Reviewer #3:

The emission reduction during the Second World Internet Conference provided a unique scenario to evaluate the chemical/physical processes affecting the air quality in Yangtze River Delta region. This paper estimated the emission reduction and simulated this scenario in a reasonable way. It provides some useful insights in the air quality management in this region. One thing is missing is this paper did not show how the chemistry works during the emission reduction period. Since sulfate and nitrate are both secondary, how they were formed and how they were affected? How did nitrate become more significant than sulfate with and without the control measures? The role of dust emission was not paid enough attention in the discussion. There is also a big room for improvement of overall writing. This paper is not presented consistently. It gives me a feeling that this paper is written by two different people. Later part was better presented than the first half.

(1) Chemistry
We have replotted figures 9-10 and inserted more discussions regarding the chemistry changes before, during and after the regulations. See Section 3.2.1 in the revised manuscript. See follows:

[revised manuscript text omitted]

(2) Dust

We do agree that dust control should be paid enough attention in this study. The dust control is also one of the major control measures during this campaign. Most construction sites were shut down, and the paved roads were added cleaning frequencies every day during the campaign. We added more discussion to the revised manuscript, as follows:

Page 3, Line 81-81: Specifically, the impact of measures such as management and control of coal-burning power plants, production restriction and suspension of industrial enterprises, motor vehicle limitation and work site suspension, dust control were investigated.

Page 15, Line 324-328: On average, mineral soluble irons (Ca$^{2+}$ and Mg$^{2+}$) declined 56.8% before and during the campaign under static conditions, this suggests that the suspension of construction operations which result in dust emissions and the rising frequency of rinsing and cleaning paved roads, significantly reduced dust emissions.

Page 20, Line 398: Emission reduction of PM$_{2.5}$ caused by dust control was estimated as 266.0 tons. Dust control contributed 10% to emission reductions of PM$_{2.5}$.

In conclusion part, (3) The effect of dust control measures is remarkable. During the conference, most of the construction sites in Jiaxing were suspended from operation. Measures of increasing frequency for road cleaning activities greatly lowered the dust emissions. Speciation of the measured PM$_{2.5}$ suggest that the mass concentration of crust material, which is greatly affected by dust, decreased by 14% compared to measurements after the conference. Specially, under static conditions, mineral soluble irons (Ca$^{2+}$ and Mg$^{2+}$) declined 56.8% before and during the campaign. This suggests that the suspension of construction operations which result in dust emissions and the rising frequency of rinsing and cleaning paved roads, significantly reduced dust emissions.

Some detail suggestions:
1. Transport vs transportation
Better not to use 'transportation of air mass'. Transportation is for traffic related business. It's used for mobile emission. A better way is to say 'the transport of air mass' for the movement of air mass/pollutants/plumes.

We have read through the manuscript and revised improper use of "transportation" to "transport" after careful check.

2. Pollution vs pollutant

The used of a lot of 'pollution' in this paper is quite confusing. I think you refer it as either 'plumes' or 'polluted air masses'. Pollution is a status, it does not mean any subject and cannot be moved around. While the plumes or pollutants can be moved or transported. I'd strongly suggest the author to check all the wordings in this paper.

We have read through the manuscript and revised improper usage of "pollution" to "plumes", "polluted air masses" or "emissions".

3. P3, line 69-70, 'Many studies: : :', 'Some have reported : : :'. Any references?

We have inserted the references, see follows:

Many studies have provided descriptive analysis of changing concentrations of air pollutants during mega events; some have reported the emission reductions and related air quality changes (Wang, et al., 2009; Wang, et al., 2010; Liu, et al., 2013; Tang, et al., 2015; Li, et al., 2016; Wang, et al.,2016; Sun, et al., 2016; Wang, et al., 2015; Chen, et al., 2017; Han, et al., 2016; Qi, et al., 2016).

4. P3, Figure 12 may be better shown here in the introduction.

We agree that putting figure 12 into the introduction part is more suitable, so we moved it forward, and revised the numbers of the figure captions accordingly.

5. P4, line 101-102, 'online' and 'On-line'?

Revised.

6. P4, line 108, 'consisting of' to 'such as/including'?

Revised.

7. P4, line 110, 'data conform' to 'data quality conform'?

Revised.

8. P5, line 137, 'with observation data and meteorological data included'. Did you used met observations for TrajStat? How?

Yes. We applied TrajStat to analyze potential source contribution areas of $PM_{2.5}$ in Jiaxing during different pollution episodes. We included observation data and meteorological data as well. For the meteorological data, we combined Global Data Assimilation System (GDAS) meteorological data provided by the NCEP (National Center for Environmental Prediction). For observation data, we included the observed hourly $PM_{2.5}$ concentrations. The long-term measurement data could be assigned to their corresponding trajectories. The model can be used to identify the trajectories to which a user can distinguish the polluted trajectories with high measurement concentration from a large number of trajectories and then the pollutant pathway could be roughly estimated. The mean pollutant concentration for each cluster can be computed using the cluster

statistics function. Pollutant pathways could then be associated with the high concentration clusters. After calculating the PSCF and CWT value, an arbitrary weight function (Polissar et al., 1999) is applied to reduce the uncertainty of cells with few endpoints. Then the potential source regions with high PSCF or CWT value could be identified. (Wang et al., 2009.) We also added color scale to PM2.5 concentrations in figures 4-8 (c).

Ref.

Polissar A V, Hopke P K, Paatero P, et al. The aerosol at Barrow, Alaska: long-term trends and source locations. Atmospheric Environment, 1999, 33(16): 2441-2458.

Wang Y Q, Zhang X Y, Draxler R R. TrajStat: GIS-based software that uses various trajectory statistical analysis methods to identify potential sources from long-term air pollution measurement data. Environmental Modelling and Software, 2009, 24(8): 938-939.

9. P5, line 140, 1x1 degree is quite coarse. Why not just used WRF simulations?

We used GDAS as the meteorological data input, these data are global assimilation data, which can well reflect the meteorological conditions and trajectories. Since we focus on the potential source regions instead of specific sources or each city, we believe 1x1 degree data should suffice for this analysis.

10. P5, line 144, 'increase with the raise of distance' to 'increase with the distance' that's true, dust PM2.5 would be the most important equal to or after sulfate. If the dust can be controlled, it's more than what has been achieved due to the control measures.

Any idea what can be done to reduce the dust emissions?

We agree that the dust control is of great importance to improve the air quality. We have highlighted the importance of dust controls, as answered in the following question 33.

The control of dust pollution includes: Construction work sites were suspended in key areas and control areas. Dust materials were forbidden to be transported within key neighbourhoods. Dust control measures were implemented on renovation operations at ports, docks, railway stations and commercial concrete mixing stations and on materials storage yards. These measures have resulted in the decrease of particle emissions and decrease of mineral ions. Speciation of the measured $PM_{2.5}$ suggest that the mass concentration of crust material, which is greatly affected by dust, decreased by 14% compared to measurements after the conference. Specially, under static conditions, mineral soluble irons ($Ca^{2+}$ and $Mg^{2+}$) declined 56.8% before and during the campaign.

28. P20, Line 393, One more evidence of other components is 33%

The original sentence "The major chemical components during this cleaner period were organic carbon (26%), nitrate (16%), ammonium (12%) and sulphate (9%)…" has been revised to "The major chemical components during this cleaner period were organic carbon (26%), nitrate (16%), ammonium (12%), sulphate (9%) and other components (37%)…".

29. P20, section 3.3.1. This section can be more concise. If needed, Details can be moved into supplement materials. The focus here is the Table3.

We agree that the section 3.3.2 and Table 4 is the major focus, so we deleted section 3.3.1, and just add a short description at the beginning of 3.3.2, which has currently been revised to 3.3.

3.3 Emissions reduction estimation during the campaign

The air quality assurance campaign for the 2nd World Internet Conference was from December 8 to December 18. In order to ensure the air quality during the conference, three provinces and Shanghai municipality in the YRD region carried out joint control measures. Based on the implementation of control measures in all areas during the conference and whether each area had effectively implemented control measures on December 8-18, regional emission reductions have been assessed…….

30. P20, line 394, 'obvious regional pollution characteristics', what is it?

It means regional transport, to avoid misunderstanding, we revised this sentence to:

The major chemical components during this cleaner period were organic carbon (26%), nitrate (16%), ammonium (12%), sulphate (9%) and other components (37%), with some newly formed particles and no obvious regional transport, suggesting that air pollutants were mainly derived from local emissions.

31. P28, line 589, 'percent reduction' to 'percentage reduction', 'conducted' to 'considered/investigated/discussed/etc'

Revised accordingly.

32. P30, section 3.6 seems to be not that relevant here. It may be moved into the introduction or the supplement.

We removed section 3.6, and revised to short descriptions in the introduction part.

Many studies have provided descriptive analysis of the changing concentrations of air pollutants during mega events, some have reported the emission reductions and related air quality changes (Wang, et al., 2009; Wang, et al., 2010; Liu, et al., 2013; Tang, et al., 2015; Li, et al., 2016; Wang, et al.,2016; Sun, et al., 2016; Wang, et al., 2015; Chen, et al., 2017; Han, et al., 2016; Qi, et al., 2016). However, different air pollution control targets, different control measures, and different locations, may cause big different effects among those strategies….

33. P32, line 682.'The effect of dust control measures is remarkable'. This conclusion comes from nowhere. It has not been discussed or showed in this paper. Better to prove it or remove it.

We revised the conclusion by adding more proves, as follows:

The effect of dust control measures is remarkable. During the conference, most of the construction sites in Jiaxing were suspended from operation. Measures of increasing frequency for road cleaning activities greatly lowered the dust emissions. Speciation of the measured $PM_{2.5}$ suggest that the mass concentration of crust material, which is greatly

affected by dust, decreased by 14% compared to measurements after the conference. Specially, under static conditions, mineral soluble irons ($Ca^{2+}$ and $Mg^{2+}$) declined 56.8% before and during the campaign. This suggests that the suspension of construction operations which result in dust emissions and the rising frequency of rinsing and cleaning paved roads, significantly reduced dust emissions.

---

## Referee Report (RR1)

Review for ACP-2018-860

The manuscript investigates the effectiveness of the emission control to the pollutant concentrations during the 2nd World Internet Conference from December 16 to December 18, 2015. The authors have addressed reviewers' points and make recommended changes in the manuscripts. The manuscript is well-organized. I suggest this manuscript goes through minor revision with the following comments.

Minor points:
1. How do you conduct the emission inventory to account for the controlled measures?

2. What is the difference between your findings and previous studies?

3. Line 106: "big different effects" to "significantly different effects"

4. Line 423: "The chemistry also changes …" , do you mean the "chemical composition of PM2.5 also changes.." or "chemical processes associated PM2.5 production".

---

## Author Response (AR2)

*Author Comments: Response to reviewers' comments*

Title: Evaluation on the effect of regional joint control measures in changing photochemical transformation: A comprehensive study of the optimization scenario analysis

Reviewer #1:

This manuscript investigated the effects of joint local and regional regulations on air pollution during the 2nd World Internet Conference held in Jiaxing, Zhejiang. Both modeling and measurements were used for the evaluation. The authors performed careful case studies by controlling the meteorological conditions, air mass back trajectory, etc. Different emission reduction plans were proposed based on different scenarios. In particular, it is recommended to implement regulation along the transport channel to the receptor-site. This is an important study to develop effective control strategies to mitigate air pollution in China. Overall, the manuscript is well-written and the analysis is solid. I recommend publication after minor revision.

Thanks to the reviewer for the comments. We have carefully revised the manuscript accordingly.

Comments:

1. Line 202. Please show the equations to calculate the metrics. Also, "Index of Agreement" should be as "IOA".

Revised. The equations have been added to the manuscript, as follows. The "I" has been revised to IOA.

Changes in manuscript:

The equations to calculate these statistical indexes are as follows:

$$NMB = \frac{\sum(P_j - O_j)}{\sum O_j} \times 100\% \tag{1}$$

$$NME = \frac{\sum |P_j - O_j|}{\sum O_j} \times 100\% \qquad (2)$$

$$IOA = 1 - \frac{\sum(P_j - O_j)^2}{\sum(|P_j - \bar{O}| + |O_j - \bar{O}|)^2} \qquad (3)$$

where $P_j$ and $O_j$ are predicted and observed hourly concentrations, respectively. $\bar{O}$ is the average value of observations. IOA ranges from 0 to 1, with 1 indicating perfect agreement between model and observation.

2.  Figure 2. Please include NMB, NME, and IOA (Table 2) in the figure.

Revised accordingly.

Changes in manuscript:

[Figure]

Fig. 3 Scatter plot of the simulated and observed PM$_{2.5}$ at the Shanxi supersite

3.  Line 241-244. This sentence has grammatical error.

Revised.

Original sentence: For each of these processes, this study has comprehensively in the integrated emission-measurement-modeling method considered the backward air flow trajectory, potential contribution source areas, meteorological conditions and the variation of $PM_{2.5}$ concentration to analyse the evolution of the observed air quality."

Changes in manuscript:

For each of these processes, this study utilized the integrated emission-measurement-modeling method to analyze the evolution of air quality from several aspects, including the backward air flow trajectory, potential source contribution areas, meteorological conditions and the variation of $PM_{2.5}$ concentration.

4. Figure 3-7. In panel (d), please specific if the PM2.5 time series is from modeling or measurement.

It is from measurement. The original sentences "(d) $PM_{2.5}$ time series for selected sites during … " have been revised to "(d) Observed $PM_{2.5}$ time series for selected sites during …" in Fig.3-7 (Fig.4-8 in the revised manuscript).

5. Figure 8 and Figure 9. These two figures are really intriguing. Why is "[SO2] after control" is similar to "[SO2 during control]", but "[SO4] after control" is much higher than "[SO4 during control]"? The opposite trend is observed for [NO2] and [NO3]. Please make similar figure for the [SO2]+[SO4] and [NO2]+[NO3], which should better represent the effect of regulation. Another potential plot is the partitioning of SOx and NOx (e.g., SO2/(SO2+SO4)). Interesting chemistry may be inferred from these analyses. Also, can the model reproduce these observations? Last comment, please consider to change the x-axis label from dates to "before/during/after regulation".

It is a very good question and suggestion!

As is shown from the figure, the $SO_2$ concentration after control is a little bit higher than during control (+5.9%). However, the $SO_4^{2-}$ after control is much higher than during control (+25.8%). This is probably due to two reasons: firstly, $SO_2$ emissions and primary sulfate emissions increased after the control measures were terminated; secondly, previous studies have reported that increased $NO_2$ emissions could accelerate the formation of secondary sulfate (Cheng et al., 2016). This can be clearly seen from the SOR. A different trend is observed for $NO_2$ and $NO_3^-$, with the $NO_2$ concentrations after control being much higher than during control (+9.4%), while the increase of $NO_3^-$ (+9.45%) is about the same. Sulfate originates from both primary emissions and secondary formation, but nitrate is mostly secondary. The NOR during and after regulation is about the same and most of the N is in the gas phase as indicated by $NOx/(NOx+NO_3^-)$ (0.87). Therefore, the increase of $NO_3^-$ is smaller than $SO_4^{2-}$. The $PM_{2.5}$ concentration after control sharply rebounded by 31.8%, indicating that both primary emissions and secondary formation are activated.

To better illustrate emissions and chemistry before, during and after control measures, we revised the previous figures and added another two indicators for partitioning of SOx /NOx, and SOR/NOR.

Changes in manuscript:

[revised manuscript text omitted]

8. Effect of local emission reductions in Jiaxing and Figure 18. Regional control only has slight extra benefit over local control. Does it suggest that less strict regulation should be implemented in nearby cities?

Figure 18 shows the decline ratio of daily average PM$_{2.5}$ concentrations under the regional emission reduction scenario, the Jiaxing local emission reduction scenario and the transport channel emission reduction scenario (24 hrs in advance and 48 hrs in advance). Air quality improvement due to regional emission reductions was slightly larger than that of local emission reductions in Jiaxing, and smaller than that of channel emission reductions. This suggests that emissions reduction in downwind cities does not have much effect on Jiaxing's air quality. In contrast, emissions reduction based on predicted transport pathway in advance are more effective than local emissions reduction.

Changes in manuscript:

In this study, the main transport channel involved is the northwest transport channel in control areas, which basically represents the typical winter transport channel in the region. Air quality improvement due to regional emission reductions was slightly larger than that of local emission reductions in Jiaxing, and smaller than that of channel emission reductions. This suggests that emissions reduction in the downwind cities does not have much effect on Jiaxing's air quality. In contrast, emissions reduction based on predicted transport pathway in advance are much more effective than local emissions reduction as well as regional emission reductions. Therefore, a well-designed management plan for the main transport channel is necessary to ensure optimized air quality improvement in autumn and winter, in addition to reducing local emissions.

*Author Comments: Response to reviewers' comments*

Title: Evaluation on the effect of regional joint control measures in changing photochemical transformation: A comprehensive study of the optimization scenario analysis

Reviewer #2:

Review for "Evaluation on the effect of regional joint control measures in changing photochemical transformation: A comprehensive study of the optimization scenario analysis"

This paper investigates the effect of regional control during the 2nd World Internet Conference from December 16 to December 18, 2015. They analyzed the meteorology condition, observed air pollutant concentration, and quantified the effect of air pollution control using numerical models. They found the local emission reduction plays an important role in air quality improvement and suggest that a 48-hr advance pollution channel control before the event. Overall, this paper is well-organized and fits into the scope of Atmospheric Chemistry and Physics on the advance understanding of atmospheric chemistry process. I suggest this paper gets accepted with the following minor revisions.

Thanks to the reviewer for the comments, we have carefully revised the manuscript accordingly.

Minor comments:

1. In the model performance section, the author mentioned about the underestimation of the simulated $PM_{2.5}$ concentration compared to the observation. Where are the uncertainties possibly coming from? Knowing this uncertainty in the model, how do we interpret the results (possible uncertainty and limitation in the result)?

We agree with the reviewer. The predicted $PM_{2.5}$ is relatively lower than the observed data (NMB values are all negative), as described in the model performance section. These underestimations may be due to three reasons. Firstly, winter underestimation of PM$_{2.5}$ (especially SOA) is a common issue with CMAQ or CAMx simulations over China (Hu et al., 2017; Li et al., 2016), which can be explained by a lack of model calculated oxidants or missing reactions (Kasibhatla et al., 1997) of SOA formation pathways (Appel et al., 2008; Foley et al., 2010; Chen et al., 2017). Secondly, uncertainty still exists in the regional emissions inventory, including the basic emissions inventory and the control scenarios as well. Thirdly, the wind speed is slightly overestimated over the region, with NMB and NME of 28% and 33%, causing fast dispersion of air pollutants and lower prediction of ambient PM$_{2.5}$ concentrations.

In view of these uncertainties, we mainly use observational data to interpret the photochemical change, while in Section 3.4, we should keep in mind that the secondary formation may probably be underestimated, causing the decline ratio lower than observed values.

Changes in manuscript:

Text has been added to interpret the model performance and the predicted results in the model performance section 2.3.2 and section 3.4.1.

Section 2.3.2

Figure 3 compares the simulated and observed PM$_{2.5}$ concentrations at the Shanxi supersite. In general, model predicted data are lower than the observed data with the NMB value of -22% to -30%, the NME value of 45% to 47% and the IOA value of 0.67 to 0.70 (Table 2). These underestimations may be due to three reasons: Firstly, winter underestimation of PM$_{2.5}$ (especially SOA) is a common issue with CMAQ or CAMx simulations over China (Hu et al., 2017; Li et al., 2016), which can be explained by a lack of model calculated oxidants or missing reactions (Kasibhatla et al., 1997) of SOA formation pathways (Appel et al., 2008; Foley et al., 2010). Secondly, uncertainty still exists in the regional emission inventory, including the basic emissions inventory and the control scenarios. Thirdly, the wind speed is slightly overestimated over the region, with

NMB and NME of 28% and 33%, causing fast dispersion of air pollutants. Overall, these statistics for both the meteorological parameters and simulated $PM_{2.5}$ are generally consistent with the results in other published modelling studies(Zheng et al., 2015;Wang et al., 2014;Zhang et al., 2011;Fu et al., 2016;Li et al., 2015b;Li et al., 2015a), which suggests that the simulation performance is acceptable.

Section 3.4.1

In view of the uncertainties of model performance (underestimation of $PM_{2.5}$, especially underestimation of SOA) described in previous sections, it is noted that the secondary formation may be underestimated, causing the decline ratio lower than reactivity.

Added references:

Appel, K.W., Bhave, P.V., Gilliland, A.B., Sarwar, G., Roselle, S.J., 2008. Evaluation of the community multiscale air quality (CMAQ) model version 4.5: sensitivities impacting model performance; part II particulate matter. Atmos. Environ. 42, 6057-6066.

Chen, Q., Fu, T. M., Hu, J., Ying, Q., & Zhang, L. (2017). Modelling secondary organic aerosols in China. National Science Review, 4(6), 806-809.

Foley, K.M., Roselle, S.J., Appel, K.W., Bhave, P.V., Pleim, J.E., Otte, T.L., Mathur, R., Sarwar, G., Young, J.O., Gilliam, R.C., Nolte, C.G., Kelly, J.T., Gilliland, A.B., Bash, J.O., 2010. Incremental testing of the community multiscale air quality (CMAQ) modeling system version, 4.7. Geosci. Model Dev. 3, 205-226.

Kasibhatla, P., Chameides, W.L., Jonn, J.S., 1997. A three dimensional global model investigation of seasonal variations in the atmospheric burden of anthropogenic sulphate aerosols. J. Geophys. Res. 102, 3737-3759.

Li, J. L., ZHANG, M. G., GAO, Y., & CHEN, L. (2016). Model analysis of secondary organic aerosol over China with a regional air quality modeling system (RAMS-CMAQ). Atmospheric and Oceanic Science Letters, 9(6), 443-450.

Hu, J., Wang, P., Ying, Q., Zhang, H., Chen, J., Ge, X., ... & Zhao, Y. (2017). Modeling biogenic and anthropogenic secondary organic aerosol in China. Atmospheric Chemistry and Physics, 17(1),

77-92.

2. Some of the figure (Figure 3-7) contents are hard to read, for example, the values on the color bar on the panel (b) and contours on the synoptic maps (a). Moreover, the graph resolution is not consistent in these Figures, especially figure (c). What is the color scale in (c)?

We have revised these figures for better visualization (Figure 4-8 in the revised manuscript). We also added the color scale to figures (c).

Changes in manuscript:

[Figure]

(a)

(b)

[Figure]

Fig. 4 Analysis of (a) the large-scale weather patterns, (b) distribution of PM$_{2.5}$ concentrations, (c) potential regional sources, (d) Observed PM$_{2.5}$ time series for selected sites during December 6 to December 8, 2015

[Figure]

Fig. 5 Analysis of (a) the large-scale weather patterns, (b) distribution of PM$_{2.5}$ concentrations, (c) potential regional sources, (d) Observed PM$_{2.5}$ time series for select sites during December 10 to December 11, 2015

Fig. 6 Analysis of (a) the large-scale weather patterns, (b) distribution of PM$_{2.5}$ concentrations, (c) potential regional sources, (d) Observed PM$_{2.5}$ time series for select sites during December 14 to December 16, 2015

[Figure]

Fig. 7 Analysis of (a) the large-scale weather patterns, (b) distribution of PM$_{2.5}$ concentrations, (c) potential regional sources, (d) Observed PM$_{2.5}$ time series for select sites during December 16 to December 18, 2015

[Figure]

Fig. 8 Analysis of (a) the large-scale weather patterns, (b) distribution of PM$_{2.5}$ concentrations, (c) potential regional sources, (d) Observed PM$_{2.5}$ time series for select sites during December 20 to December 23

3. Line 153: "GDAS" needs to be defined at its first appearance.

"GDAS" has been revised to "Global Data Assimilation System (GDAS)".

4. Line 201: : : : Index of Agreement (IOA). Same apply to Line 209:and the IOA value of 0.67 to 0.70.

Revised.

5. Line 340: " under static weather condition"

Revised.

6. Figure 9: what is the unit of the measurement (%)?

The unit is "μg/m$^3$". It has been revised in the manuscript.

7. Figure 11: WS/WD panel has similar information as the $PM_{2.5}$ (top panel) regarding the wind direction. I suggest change the WS/WD panel to wind speed only and use contour lines to represent that.

The top panel with different colors indicates the trajectories at 500m height, which can be used to represent long-range transport; the WS/WD panel indicates surface wind, which can give information regarding pollution dispersion or accumulation. Therefore, we believe it is better to keep both.

8. Line 649-652: Please be consistent on the notification, such as SO2 PM2.5. This occurs in other sections of the manuscript, e.g. line 669-672. 2019.

We have gone through the manuscript and made edits accordingly.

*Author Comments: Response to reviewers' comments*

Title: Evaluation on the effect of regional joint control measures in changing photochemical transformation: A comprehensive study of the optimization scenario analysis

Reviewer #3:

The emission reduction during the Second World Internet Conference provided a unique scenario to evaluate the chemical/physical processes affecting the air quality in Yangtze River Delta region. This paper estimated the emission reduction and simulated this scenario in a reasonable way. It provides some useful insights in the air quality management in this region. One thing is missing is this paper did not show how the chemistry works during the emission reduction period. Since sulfate and nitrate are both secondary, how they were formed and how they were affected? How did nitrate become more significant than sulfate with and without the control measures? The role of dust emission was not paid enough attention in the discussion. There is also a big room for improvement of overall writing. This paper is not presented consistently. It gives me a feeling that this paper is written by two different people. Later part was better presented than the first half.

Thanks to the reviewer for the comments. We have carefully revised the manuscript accordingly, especially in providing insights into the changes of chemistry and dust impact as well. Follows are detailed discussions and revisions.

● Chemistry

To get insights into the changes of chemistry, we replotted figures 9-10, added SOR/NOR/partition of gas phase vs gas + particle phase, and added more discussions regarding the chemistry changes before, during and after the regulations (we selected static weather conditions to set aside the impact of transport). Figure 9 shows the concentration of criteria pollutants including $SO_2$, NO, CO, $NO_2$ and $PM_{2.5}$ before, during and after the regulation under stagnant weather conditions. It can be seen that pollutant concentrations during the campaign were less than those before the campaign, in which $SO_2$ had the most significant decline of 40.1%, NOx, CO, $PM_{2.5}$ and $PM_{10}$ declined 8.0%, 2.6%, 12.5% and 16.3%, respectively, indicating that control measures have significantly improved the air quality in Jiaxing City, especially in the reduction of primary emissions of $SO_2$ and $PM_{10}$.

However, after the campaign, all pollutant concentrations rebounded sharply. $SO_2$, NO, $NO_2$, CO, $PM_{2.5}$, $PM_{10}$ increased 8.3%, 15.4%, 10.3%, 31.8%, 32.2% and 28.6%, respectively. Concentrations of some pollutants were even higher than those before the campaign, suggesting that the source emission intensity had significantly increased after the campaign; the rebounding ratio of NOx is higher than $SO_2$.

The changes of major $PM_{2.5}$ chemical components before, during and after the campaign under static weather conditions, could be utilized to characterize the changes of atmospheric chemistry. The concentrations of major chemical components of $PM_{2.5}$ during the campaign were less than those before the campaign, which is consistent with the observation for criteria pollutant concentrations. On average, $SO_4^{2-}$, $NH_4^+$, $NO_3^-$, OC, mineral soluble irons ($Ca^{2+}$ and $Mg^{2+}$) and $K^+$ declined 11.8%, 5.1%, 32.1%, 9.8%, 56.8% and 5.1%, respectively. During the campaign, $NO_3^-$ significantly decreased, indicating that vehicle control measures successfully reduced $NO_x$ emissions and subsequently the formation of inorganic aerosols. Significant decrease in $SO_4^{2-}$ also indicate that restricting and/or suspending the operation of coal-burning boilers in local and neighbouring cities had a positive impact.

The chemistry also changes if we compare during and after the regulation. As is shown from figure 10, the $SO_2$ concentrations after control is a little bit higher than during control (+5.9%). However, the $SO_4^{2-}$ after control is much higher than during control (25.8%). This is probably due to two reasons: firstly, increase of $SO_2$ emissions and primary sulfate emissions after the control measures were terminated; secondly, previous studies have reported that increased $NO_2$ emissions could accelerate the formation of secondary sulfate (Cheng et al., 2016). This can be clearly seen from the SOR and NOR indicators. A different trend is observed for $NO_2$ and $NO_3^-$, with the $NO_2$ concentrations after control being much higher than during control (+9.4%), while the increase of $NO_3^-$ (+9.45%) is about the same. Sulfate originates from both primary emissions and secondary formation, but nitrate is mostly secondary. The NOR during and after regulation is about the same and most of the N is in the gas phase as indicated by $NOx/(NOx+NO_3^-)$ (0.87). Therefore, the increase of $NO_3^-$ is lower than $SO_4^{2-}$.

From the descriptions above, we can see that the secondary formation of nitrate was greatly slowed down due to the strict emission reduction measures. But after the control measures were terminated, the secondary formation rebounded, with the rebounding ratio of sulfate higher (26%) than nitrate (9%). But in terms of absolute concentrations, the nitrate is higher than sulfate, showing that nitrate has become the most important chemical species within $PM_{2.5}$ in winter in the YRD region.

[Figure]

Fig. 9 Comparison between air pollutant concentrations at Shanxi station before, during, and after the campaign under stagnant meteorological conditions

[Figure]

[Figure]

[Figure]

Fig. 10 Comparison between PM$_{2.5}$ chemical components at Shanxi station before and after the campaign under static meteorological conditions

- Dust

We do agree that dust control should be paid enough attention in this study. The dust control is also one of the major control measures during this campaign. Most construction sites were shut down, and cleaning frequencies of the paved roads were increased during the campaign. We added more discussions in the revised manuscript.

Changes in manuscript:

Page 3, Line 82: Specifically, the impact of measures such as management and control of coal-burning power plants, production restriction and suspension of industrial enterprises, motor vehicle limitation and work site suspension, dust control were investigated.

Page 16, Line 331: On average, SO$_4^{2-}$, NH$_4^+$, NO$_3^-$, OC, mineral soluble irons (Ca$^{2+}$ and Mg$^{2+}$) and K$^+$ declined 11.8%, 5.1%, 32.1%, 9.8%, 56.8% and 5.1%, respectively. Comparisons between the distribution of PM$_{2.5}$ chemical components before and during the campaign under static conditions suggest that Ca$^{2+}$ and Mg$^{2+}$ decreased most significantly during the control period, which indicates that the suspension of construction operations which result in dust emissions and the rising frequency of rinsing and cleaning paved roads, significantly reduced dust emissions.

Page 21, Line 406: Emission reduction of $PM_{2.5}$ caused by dust control was estimated to be 266.0 tons. Dust control contributed 10% to emission reductions of $PM_{2.5}$.

In the conclusion part, (3) The effect of dust control measures is remarkable. During the conference, most of the construction sites in Jiaxing were suspended from operation. Increased frequency for road cleaning activities greatly lowered the dust emissions. Speciation of the measured $PM_{2.5}$ suggest that the mass concentration of crust material, decreased by 14% compared to measurements after the conference. Specially, under static conditions, mineral soluble irons ($Ca^{2+}$ and $Mg^{2+}$) declined 56.8% before and during the campaign. This suggests that the suspension of construction operations and increased frequency of rinsing and cleaning of paved roads significantly reduced dust emissions.

● Writing

We have read through the manuscript and revised the language thoroughly.

Some detail suggestions:

1. Transport vs transportation

Better not to use 'transportation of air mass'. Transportation is for traffic related business. It's used for mobile emission. A better way is to say 'the transport of air mass' for the movement of air mass/pollutants/plumes.

We have read through the manuscript and revised improper use of "transportation" to "transport" after careful check.

2. Pollution vs pollutant

The used of a lot of 'pollution' in this paper is quite confusing. I think you refer it as either 'plumes' or 'polluted air masses'. Pollution is a status, it does not mean any subject and cannot be moved around. While the plumes or pollutants can be moved or transported. I'd strongly suggest the author to check all the wordings in this paper.

We have read through the manuscript and revised improper usage of "pollution" to "plumes", "polluted air masses" or "emissions".

3. P3, line 69-70, 'Many studies: : :', 'Some have reported : : :'. Any references?

We have inserted the references.

Changes in manuscript:

Many studies have provided descriptive analysis of changing concentrations of air pollutants during mega events; some have reported the emission reductions and related air quality changes (Wang, et al., 2009; Wang, et al., 2010; Liu, et al., 2013; Tang, et al., 2015; Li, et al., 2016; Wang, et al.,2016; Sun, et al., 2016; Wang, et al., 2015; Chen, et al., 2017; Han, et al., 2016; Qi, et al., 2016).

4. P3, Figure 12 may be better shown here in the introduction.

We agree that putting figure 12 into the introduction part is more suitable, so we moved it forward, and revised the numbers in the figure captions accordingly.

Changes in manuscript:

These areas cover 9 cities including Jiaxing, Huzhou, Hangzhou, Ningbo and Shaoxing in Zhejiang province, Suzhou and Wuxi in Jiangsu province and Xuancheng in Anhui province, as shown in Fig.1.

[Figure]

Fig.1 Controlled regions in the Action Plan for Air Quality Control during the World Internet Conference

5. P4, line 101-102, 'online' and 'On-line'?

Revised.

6. P4, line 108, 'consisting of' to 'such as/including'?

Revised.

7. P4, line 110, 'data conform' to 'data quality conform'?

Revised.

8. P5, line 137, 'with observation data and meteorological data included'. Did you used met observations for TrajStat? How?

Yes. We applied TrajStat to analyze potential source contribution areas of PM$_{2.5}$ in Jiaxing during different pollution episodes. We included observation data and meteorological data as well. For the meteorological data, we combined Global Data Assimilation System (GDAS) meteorological data provided by the NCEP (National Center for Environmental Prediction). For observation data, we included the observed hourly PM$_{2.5}$ concentrations. The long-term measurement data could be assigned to their corresponding trajectories. The model can be used to identify the trajectories to which a user can distinguish the polluted trajectories with high measurement concentration from a large number of trajectories and then the pollutant pathway could be roughly estimated. The mean pollutant concentration for each cluster can be computed using the cluster statistics function. Pollutant pathways could then be associated with the high concentration clusters. After calculating the PSCF and CWT value, an arbitrary weight function (Polissar et al., 1999) is applied to reduce the uncertainty of cells with few endpoints. Then the potential source regions with high PSCF or CWT value could be identified. (Wang et al., 2009.) We also added color scale to PM2.5 concentrations in figures 4-8 (c).

Ref.

Polissar A V, Hopke P K, Paatero P, et al. The aerosol at Barrow, Alaska: long-term trends and source locations. Atmospheric Environment, 1999, 33(16): 2441-2458.

Wang Y Q, Zhang X Y, Draxler R R. TrajStat: GIS-based software that uses various trajectory statistical analysis methods to identify potential sources from long-term air pollution measurement data. Environmental Modelling and Software, 2009, 24(8): 938-939.

9. P5, line 140, 1x1 degree is quite coarse. Why not just used WRF simulations?

We used GDAS as the meteorological data input. These data are global assimilation data, which can well reflect the meteorological conditions and trajectories. Since we focus on the potential source regions instead of specific sources or each city, we believe 1x1 degree data should suffice for this analysis.

10. P5, line 144, 'increase with the raise of distance' to 'increase with the distance' that's true, dust PM2.5 would be the most important equal to or after sulfate. If the dust can be controlled, it's more than what has been achieved due to the control measures.

Any idea what can be done to reduce the dust emissions?

We agree that the dust control is of great importance to improve the air quality. We have highlighted the importance of dust controls, as answered in the following question 33.

The control of dust pollution includes: Construction work sites were suspended in key areas and control areas. Transport of dust materials were forbidden within key neighborhoods. Dust control measures were implemented on renovation operations at ports, docks, railway stations and commercial concrete mixing stations and on materials storage yards. These measures have resulted in the decrease of particle emissions and decrease of mineral ions. Speciation of the measured $PM_{2.5}$ suggest that the mass concentration of crust material, which is greatly affected by dust, decreased by 14% compared to measurements after the conference. Specially, under static conditions, mineral soluble irons ($Ca^{2+}$ and $Mg^{2+}$) declined 56.8% before and during the campaign.

28. P20, Line 393, One more evidence of other components is 33%

The original sentence "The major chemical components during this cleaner period were organic carbon (26%), nitrate (16%), ammonium (12%) and sulphate (9%)…" has been revised to "The major chemical components during this cleaner period were organic carbon (26%), nitrate (16%), ammonium (12%), sulphate (9%) and other components (37%)…".

29. P20, section 3.3.1. This section can be more concise. If needed, Details can be moved into supplement materials. The focus here is the Table3.

We agree that the section 3.3.2 and Table 4 is the major focus, so we deleted section 3.3.1, and just add a short description at the beginning of 3.3.2, which has currently been revised to 3.3.

3.3 Emissions reduction estimation during the campaign

The air quality assurance campaign for the 2nd World Internet Conference was from December 8 to December 18. In order to ensure the air quality during the conference, three provinces and Shanghai municipality in the YRD region carried out joint control measures. Based on the implementation of control measures in all areas during the conference and whether each area had effectively implemented control measures on December 8-18, regional emission reductions have been assessed…….

30. P20, line 394, 'obvious regional pollution characteristics', what is it?

It means regional transport, to avoid misunderstanding, we revised this sentence to:

The major chemical components during this cleaner period were organic carbon (26%), nitrate (16%), ammonium (12%), sulphate (9%) and other components (37%), with some newly formed particles and no obvious regional transport, suggesting that air pollutants were mainly derived from local emissions.

31. P28, line 589, 'percent reduction' to 'percentage reduction', 'conducted' to 'considered/investigated/discussed/etc'

Revised accordingly.

32. P30, section 3.6 seems to be not that relevant here. It may be moved into the introduction or the supplement.

We removed section 3.6, and revised to short descriptions in the introduction part.

Many studies have provided descriptive analysis of the changing concentrations of air pollutants during mega events, some have reported the emission reductions and related air quality changes (Wang, et al., 2009; Wang, et al., 2010; Liu, et al., 2013; Tang, et al., 2015; Li, et al., 2016; Wang, et al.,2016; Sun, et al., 2016; Wang, et al., 2015; Chen, et al., 2017; Han, et al., 2016; Qi, et al., 2016). However, different air pollution control targets, different control measures, and different locations, may cause big different effects among those strategies….

33. P32, line 682.'The effect of dust control measures is remarkable'. This conclusion comes from nowhere. It has not been discussed or showed in this paper. Better to prove it or remove it.

We revised the conclusion by adding more proves, as follows:

[revised manuscript text omitted]

(a)

[Figure]

(a)

2015-12-07 06:00 Distribution of PM₂.₅ concentration

2015-12-07 09:00 Distribution of PM₂.₅ concentration (b)

[Figure]

(c)

(d)

[revised manuscript text omitted]

(a)

2015-12-16 09:00 Distribution of PM$_{2.5}$ concentration μg/m³

2015-12-17 12:00 Distribution of PM$_{2.5}$ concentration μg/m³

2015-12-18 10:00 Distribution of PM$_{2.5}$ concentration μg/m³

(b)

(c)

(d)

[revised manuscript text omitted]

---

## Author Response (AR3)

*Author Comments: Response to reviewers' comments*

Title: Evaluation on the effect of regional joint control measures in changing photochemical transformation: A comprehensive study of the optimization scenario analysis

The manuscript investigates the effectiveness of the emission control to the pollutant concentrations during the 2nd World Internet Conference from December 16 to December 18, 2015. The authors have addressed reviewers' points and make recommended changes in the manuscripts. The manuscript is well- organized. I suggest this manuscript goes through minor revision with the following comments.

We would like to take this opportunity to express our sincere thanks the anonymous reviewers for the comments and suggestions, which have helped greatly to improve this paper.

Minor points:

1. How do you conduct the emission inventory to account for the controlled measures?

**Firstly, we have developed a basic emission inventory for the YRD region based on a bottom**

**up methodology with activity data, emission factors and control technologies as inputs.**

**Secondly, we further developed a controlled emission inventory to account for the measures**

**based on the emission reduction requirements described in the control measures plan.**

Based on the control measures plan, three provinces and Shanghai municipality in the YRD region carried out joint control measures and three kind of areas with different emission reduction requirements were established, including (i)key areas, (ii)strict control areas, (iii)control areas and extension areas, respectively. Among them, key areas and strict control areas included Zhejiang province (including Hangzhou, Ningbo, Huzhou, Jiaxing and Shaoxing), Shanghai (including Jinshan and Fengxian), Jiangsu province (including Suzhou and Wuxi) and Anhui province (including

Xuancheng, Ma'anshan and Wuhu). The following measures were taken in the key areas and control areas: (1) Strictly control emissions from coal-burning power plants: reduce emissions from power plants which have not completed ultra-low emission transmission processes by 50% in key areas and by 30% in control areas. (2) Reduce emissions from key facilities: restriction or suspension of production were imposed on industries including cement, steel, construction materials, petrochemicals, chemicals, casting, leather, non-ferrous metals, plate glass, pharmaceuticals, surface spraying and printing. Activities from all key facilities in the key areas were suspended (maximum production limits were imposed on steel and petrochemical industries), while activities from key facilities in the control areas were reduced by 30%. Facilities which could not meet the emission standards in a stable way, or facilities that are not equipped with exhaust gas treatment or the exhaust gas treatment equipment cannot operate normally were suspended. Operation and/or maintenance at petrochemical and chemical facilities were prohibited. (3) Strictly control motor vehicle emissions: in the core areas of Zhejiang province, motor vehicle restrictions were implemented, which means that low-speed trucks were prohibited to pass except for people's livelihood-related activities. Vehicles which had not obtained valid qualifications for environmental inspection were prohibited for operation. (4) Control dust emissions: Activities at construction sites were suspended in the key areas and control areas. Dust materials were prohibited to be transported within key neighbourhoods. Dust control measures were implemented on renovation operations at ports, docks, railway stations and commercial concrete mixing stations and at materials storage yards. (5) Control emissions from other sources: in the key areas and control areas, oil storage facilities, gas stations or tank trucks which were not equipped with equipment for oil and gas recycling or equipment that could not operate normally were prohibited for trade or transporting oil products. Outdoor barbecue, burning activities in the open air were prohibited. All the primary schools, secondary schools, kindergartens, institutions and public institutions in Jiaxing were given a three-day vacation. **Based on the emission reduction ratio and suspended emission sources including both industries and fugitive emissions, we estimated the emissions reductions.**

In addition, during the campaign, some cities took stricter emission reduction measures for predicted possible upcoming severe air pollution events. Shanghai started temporary control of heavy pollution on December 14 and initiated the yellow warning for heavily polluted weather on December 15. Emergency control measures were initiated on December 15 with strengthening control efforts for industries, work sites and motor vehicles. **For these urgent control measures, we further calculated emissions reductions and consolidated into the controlled emissions inventory.**

**Based on the control measures and emission reduction requirements mentioned above, we developed a controlled emission inventory to account for the control measures.**

Changes in manuscript:

In the end of Section 2.3.1, we inserted the following descriptions regarding the emissions inventory accounting for control measures:

We further developed a controlled emission inventory to account for the control measures based on the emission reduction requirements described in the control measures paln and the control measures for the emergency air pollution warning. These estimates are basically according to the control measures and reduction requirements for specific source sectors and cities described the control plan.

2. What is the difference between your findings and previous studies?

China has successfully implemented air pollution control plans and ensured good air quality for several mega events. After implementation of these control measures, it is important to understand how effective these strategies are. Previous studies mainly reported the effects of those control measures on the air quality based on observations, emissions or simulations. For example, Lu et al (2016) reported the influence of control measures on air quality during the second Asian Youth Games in Nanjing based on measurements of 15 trace metal elements. Tang et al (2015) reported the impact of emission controls on air quality in Beijing during APEC based on lidar ceilometer observations. Liu et al., (2013) reported the emission controls and changes in air quality in Guangzhou during the Asian Games based on combination of observation data, emission reduction measures and air quality simulations. Han et al., (2016) reported the effect of the pollution control measures on $PM_{2.5}$ during the 2015 China Victory Day Parade based on measurements, and found that the decrease in concentration of water-soluble ions in $PM_{2.5}$, which results from variations in air mass transport. Wang et al., (2016) reported the relative impact of emissions controls and meteorology on air pollution mitigation associated with the Asia-Pacific Economic Cooperation (APEC) conference in Beijing, China; they used PMF to analyse changes of the sources. Wang et al., (2015) summarized the atmospheric composition changes before and during the APEC period for four stages from 20 October to 20 November 2014; they also assessed the change in atmospheric composition after the implementation of various control measures on a regional scale. Of the previous studies, changes of chemistry before, during and after these mega events are seldom discussed. This study reports the changes of compositions and chemistry from these combined analyses. These are the main difference. Since we do not have similar or different results to compare, we did not insert these contents into the manuscript.

By saying "The chemistry also changes…", we mean both the chemical compositions of $PM_{2.5}$ and chemical processes associated with $PM_{2.5}$ production change, which can be shown by the following interpretations including changes of concentrations and the difference of SNA formation.

Changes in manuscript:

[revised manuscript text omitted]